# Cancer cells copy migratory behavior and exchange signaling networks via extracellular vesicles

Sander C Steenbeek[1,2], Thang V Pham[3], Joep de Ligt[4], Anoek Zomer[2], Jaco C Knol[3], Sander R Piersma[3], Tim Schelfhorst[3], Rick Huisjes[5], Raymond M Schiffelers[5], Edwin Cuppen[4] (iD), Connie R Jimenez[3,*,†] (iD) & Jacco van Rheenen[1,2,†,**] (iD)

## Abstract

Recent data showed that cancer cells from different tumor subtypes with distinct metastatic potential influence each other's metastatic behavior by exchanging biomolecules through extracellular vesicles (EVs). However, it is debated how small amounts of cargo can mediate this effect, especially in tumors where all cells are from one subtype, and only subtle molecular differences drive metastatic heterogeneity. To study this, we have characterized the content of EVs shed *in vivo* by two clones of melanoma (B16) tumors with distinct metastatic potential. Using the Cre-LoxP system and intravital microscopy, we show that cells from these distinct clones phenocopy their migratory behavior through EV exchange. By tandem mass spectrometry and RNA sequencing, we show that EVs shed by these clones into the tumor microenvironment contain thousands of different proteins and RNAs, and many of these biomolecules are from interconnected signaling networks involved in cellular processes such as migration. Thus, EVs contain numerous proteins and RNAs and act on recipient cells by invoking a multi-faceted biological response including cell migration.

**Keywords** Cre-LoxP; extracellular vesicles; intratumoral heterogeneity intravital microscopy; signaling networks
**Subject Categories** Cancer; Membrane & Intracellular Transport; Post-translational Modifications, Proteolysis & Proteomics
**The EMBO Journal (2018) 37: e98357**

## Introduction

Tumors are heterogeneous with respect to mutational pattern, gene expression, protein profiles, and microenvironment (e.g., oxygen and nutrient levels, extracellular matrix composition, and immune cell infiltration; Hamm *et al*, 2010; Swanton, 2012; Junttila & de Sauvage, 2013; Venkatesan & Swanton, 2016). This heterogeneity leads to large functional intratumoral differences in cancer cell behavior so that a tumor may contain or develop small numbers of cells that possess properties to migrate, metastasize, or evade therapy treatment (Scheele *et al*, 2016). Therefore, intratumoral heterogeneity has directly been linked to disease progression and metastasis (Inda *et al*, 2010; Calbo *et al*, 2011; Cleary *et al*, 2014) and has been suggested to be the main reason for cancer treatment failure in the clinic (Burrell & Swanton, 2014; Jamal-Hanjani *et al*, 2015).

In recent years, extracellular vesicles (EVs) have attracted a lot of attention as microenvironmental intercellular messengers that may severely complicate tumor heterogeneity (Zomer & van Rheenen, 2016). EVs are small lipid membrane-enclosed vesicles that carry biologically active molecules including lipids, proteins, DNA, and various RNA species such as mRNA, miRNA, and lncRNA. A variety of EVs are described, including exosomes arising from fusion of multivesicular bodies (MVBs) with the limiting plasma membrane, shed microvesicles or large oncosomes budding from the limiting plasma membrane, and apoptotic bodies being released by apoptotic cells (Raposo & Stoorvogel, 2013). Because it remains to be determined what the contribution of these populations is in the *in vivo* tumor microenvironment, we use the collective term "extracellular vesicles" to commonly refer to all EV subtypes (Gould & Raposo, 2013). EV-associated biomolecules such as EV-RNA are stable in EVs and functional upon delivery into recipient cells. For example, upon EV uptake, vesicular mRNA is translated into functional proteins (Valadi *et al*, 2007), and vesicular miRNAs suppress target genes in recipient cells (Hergenreider *et al*, 2012; Fong *et al*, 2015; Zhang *et al*, 2015). Moreover, EV-RNA-based reporter systems have confirmed the transport of functional mRNA for Cre (Ridder *et al*, 2015; Zomer *et al*, 2015, 2016) or GlucB (Lai *et al*, 2015) into recipient cells. Thus, by exchanging EVs, cells can transfer biomolecules to recipient cells, thereby

1   Molecular Pathology, Oncode Institute, The Netherlands Cancer Institute, Amsterdam, The Netherlands
2   Oncode Institute, Hubrecht Institute-KNAW & University Medical Centre Utrecht, Utrecht, The Netherlands
3   OncoProteomics Laboratory, Department of Medical Oncology, Cancer Center Amsterdam, VU University Medical Center, Amsterdam, The Netherlands
4   Division Biomedical Genetics, Center for Molecular Medicine, Oncode Institute, University Medical Centre Utrecht, Utrecht, The Netherlands
5   Department of Clinical Chemistry and Haematology, University Medical Center Utrecht, Utrecht, The Netherlands
    *Corresponding author. Tel: +31 20 4442340; E-mail: c.jimenez@vumc.nl
    **Corresponding author. Tel: +31 20 5126906; E-mail: j.v.rheenen@nki.nl
    †These authors contributed equally to this work

potentially influencing the recipient cell's behavior (Valadi *et al*, 2007) and tumor heterogeneity (Zomer & van Rheenen, 2016). For instance, EV-mediated crosstalk between cancer cells and non-cancer cells has been linked to promote tumor growth by inducing proliferation (Rajappa *et al*, 2016; Richards *et al*, 2017) and angiogenesis (Park *et al*, 2010; Umezu *et al*, 2014; Feng *et al*, 2017). In addition, cancer cell-derived EVs have been shown to promote metastasis by priming of the pre-metastatic niche (Costa-Silva *et al*, 2015; Hoshino *et al*, 2015; Liu *et al*, 2016) and inducing leakiness of the blood–brain barrier (Tominaga *et al*, 2015; Treps *et al*, 2016). However, in addition to the non-transformed cell types, cancer cells are also highly exposed to cancer cell-released EVs. Importantly, recent data suggest that cancer cells can also take up cancer EVs. EV exchange between cancer cell subsets with different phenotypic properties has been shown to transfer apoptosis resistance (Wojtuszkiewicz *et al*, 2016) and drug resistance (Sousa *et al*, 2015), and metastatic (Zomer *et al*, 2015) properties.

Although accumulating evidence suggests that cancer cells can phenocopy behavior through exchange of EVs, it is widely debated how the transfer of small amounts of cargo can mediate this effect. For example, it is hard to imagine how transfer of single proteins or RNAs can induce a phenotypic change. In a proof of concept study, we have previously shown that the highly aggressive basal breast cancer cell line MDA-MB-231 can phenocopy its metastatic behavior to the more benign luminal A breast cancer line T47D when co-transplanted into immune-deficient mice (Zomer *et al*, 2015). Although these proof-of-principle experiments illustrate the potential of EV exchange to phenocopy differential behavior, they do not model EV exchange in tumors with cells from the same subtype. To model the more physiological and moderate subclonal heterogeneity, we here study the exchange of EV content between two syngenic melanoma cancer lines with distinct metastatic potential: the B16F1 and B16F10 model (Hart & Fidler, 1980; Poste *et al*, 1980; Cillo *et al*, 1987; Nakamura *et al*, 2002; Mathieu *et al*, 2007). Importantly, these cancer cell lines have a common origin, and therefore, differences in phenotype and EV cargo are linked to clonal disease progression and developed metastatic potential, instead of tumor origin or subtype. B16F10 cells have been shown to shed EVs that can educate bone marrow-derived cells (Peinado *et al*, 2012), but it is unknown to what extent these vesicles influence cancer subclones with a less metastatic phenotype. Here, we studied the mutual influence of cancer subclones and showed large discrepancy between the efficiency of EV transfer *in vitro* and *in vivo,* underlining the importance of studying EV exchange between cells in their *in vivo* setting. We isolated EVs from the *in vivo* setting and identified that cancer cell subclones with distinct metastatic potential transfer RNAs and proteins that are interconnected in networks involved in migration, leading to phenocopying of migratory behavior.

## Results and Discussion

### Modeling tumor heterogeneity using the B16F1 and B16F10 model

To investigate the influence of EVs on heterogeneity of cancer cell behavior, we studied two clones that were derived from serial transplantations of a melanoma (B16) that developed spontaneously behind the ear of a C57BL/6 mouse (El, 1962). These clones, B16F1 and B16F10, have been shown to have differential metastatic potential, with the B16F10 model being more metastatic than the B16F1 model upon intravenous injection of cancer cells (Hart & Fidler, 1980; Poste *et al*, 1980; Cillo *et al*, 1987; Nakamura *et al*, 2002; Mathieu *et al*, 2007). Subcutaneous injection of fluorescently labeled B16F1 and B16F10 cells in 20 C57BL/6 mice led to tumors within 3–4 weeks, and co-injection of both lines led to tumors that contain both fluorescent B16 subclones. In nine mice, B16F10 cells formed the majority (> 50%) of the cancer cells whilst in 11 mice, the majority was formed by B16F1 cancer cells. Examination of lungs, lymph nodes, and livers showed the presence of micrometastases derived from B16F10 cells (six out of nine mice in which B16F10 cells are the major cancer type) but only occasionally from B16F1 cells (one out of 11 mice in which the B16F1 cells are the major cancer type) confirming their differential metastatic potential. Next, we tested whether we observe differential behavior in an early step of metastasis, namely migration. Using intravital microscopy (IVM), we tracked the migration of individual cells within one imaging field (Fig 1A and B), and averaged the migration speed of each cell type, connecting measurements from the same imaging field with a line (Fig 1C). Consistent with other studies, we observe that average cancer cell migration speed varies between imaging fields, as previously proposed most likely due to microenvironmental differences (Condeelis & Segall, 2003; Joyce & Pollard, 2009; Zomer *et al*, 2015). In heterogeneous tumors, B16F10 cancer cells have an average 1.5-fold higher migration speed than B16F1 cancer cells present in the same tumor area, confirming that cells demonstrated to be more metastatic have a higher migration speed (Fig 1D). From these results, we conclude that co-injection of B16F1 and B16F10 cells leads to tumors that consist of cancer cells with heterogeneous metastatic behavior.

### Cancer cells functionally exchange EVs *in vivo*

Next, we studied whether the B16F1 and B16F10 cells release and exchange EVs. First, EVs were purified from *in vitro* cultures using ultracentrifugation and stained with the lipophilic dye PKH67. To test whether B16F1 cells can take up EVs released from B16F10 cells and vice versa, we added labeled EVs to recipient cells of the other cell type. We observed that the pool of EVs enriched at a lower centrifugation speed (16,500 *g*) and the pool of EVs enriched at a higher speed (100,000 *g*; Thery *et al*, 2006; Greening *et al*, 2015; Szatanek *et al*, 2015) are both taken up by recipient cells *in vitro* (Fig 1E). To test whether the mutual uptake of EVs also led to the functional release of the content in the recipient cells, we employed the Cre-LoxP system (Ridder *et al*, 2015; Zomer *et al*, 2015, 2016). In the EVs released by Cre-expressing B16 cells, the mRNA of Cre was present (Fig 1F). Reporter cells that take up these EVs, and get exposed to the luminal cargo, switch expression of DsRed to eGFP (Fig 1G). Indeed, we observed reporter[+] cells that report Cre activity only in tumors that also contained Cre[+] cells (Fig 1H and J). Importantly, GFP[+] cells did not express CFP, excluding that the reporter[+] cells fused with Cre[+] cells (Fig EV1A–D). These data suggest that the color switch reports EV-mediated functional transfer of Cre activity. Nevertheless, formally we cannot exclude a small fraction of the Cre exchange may have occurred via EV-independent mechanisms, such as Cre transfer through cell–cell contacts.

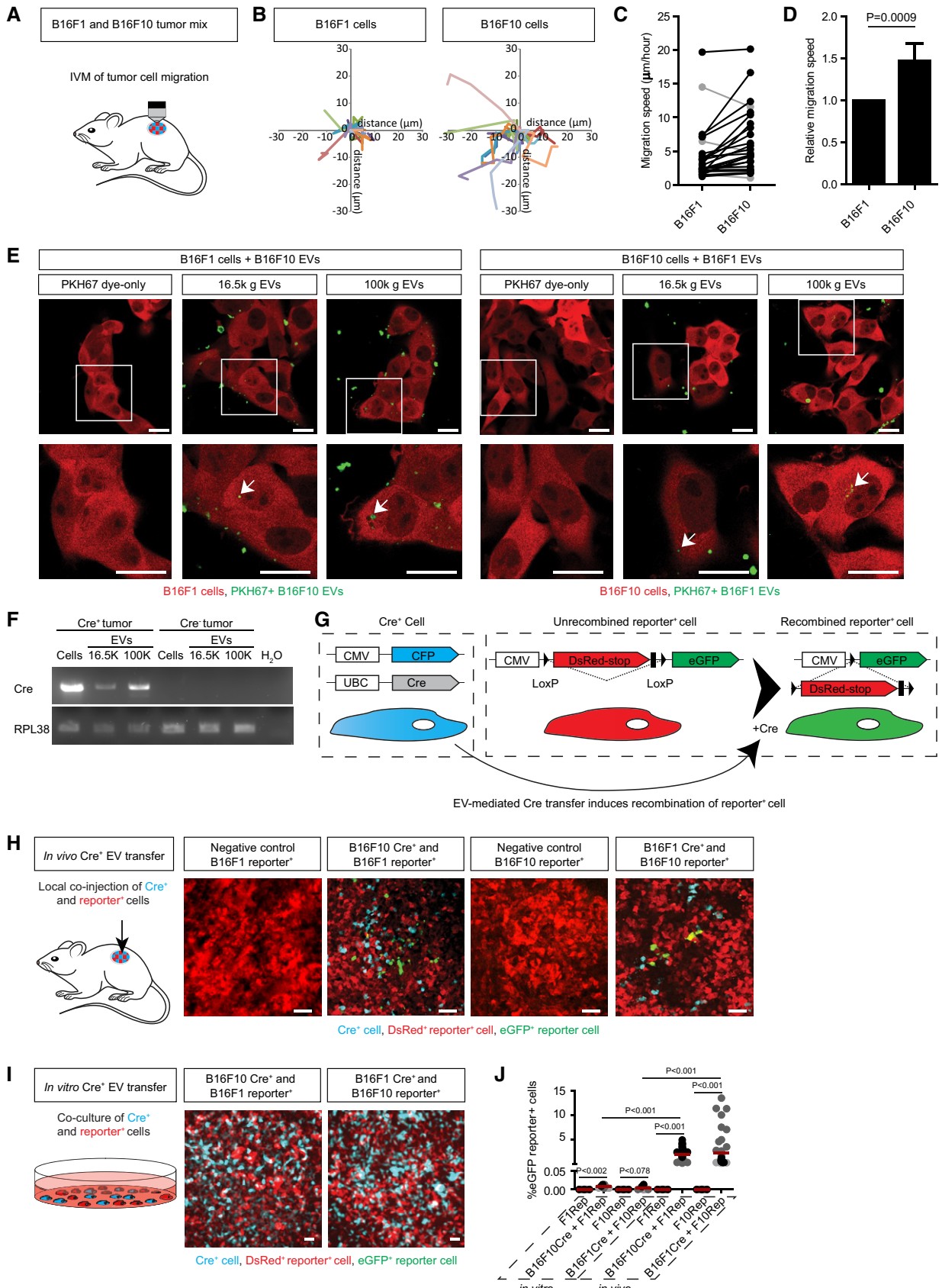

**Figure 1.**

**Figure 1.  Melanoma subclones with differential phenotypes functionally exchange EVs.**

A  Cartoon displaying intravital microscopy to study tumor cell migration in B16F1 and B16F10 mixed tumors.
B  Representative rose plots of tumor cell migration tracks of B16F1 and B16F10 tumor cells within the same imaging field, track shown for 2h30 migration.
C  Average migration speed of B16F1 and B16F10 tumor cells within the same imaging field is connected with a line. Black lines show faster migration in B16F10 cells, and gray lines show faster migration of B16F1 cells within one imaging field. B16F10 cells have an average faster migration speed in 22 of 26 positions.
D  Relative migration speed of B16F10 cells to the average B16F1 cell migration speed. Data represented as mean ± SEM with the Wilcoxon signed rank test. $n = 26$ positions in eight mice.
E  Single z plane optical sections confirm uptake of *in vitro*-derived 16.5K and 100K EVs across cell types 3 h after addition to culture medium. B16F10-derived EVs were added to B16F1 cells (left) and vice versa (right), arrows point to internalized EVs, scale bar 20 μm.
F  Cells, 16.5K EVs, and 100K EVs were isolated from B16F10 Cre[+] and B16F1 Cre[−] tumors. RT-PCR for Cre and ribosomal protein L38 (RPL38) mRNA was performed as indicated.
G  Cartoon of the Cre-LoxP system to study functional Cre[+] EV transfer. Reporter[+] cells change fluorophore expression upon uptake of Cre[+] EVs from the Cre[+] donor cells followed by excision of the DsRed-stop.
H  Cartoon and representative images of reporter[+] only tumors and local *in vivo* Cre[+] and reporter[+] B16F1 and B16F10 tumor mixes, scale bar 50 μm.
I  Cartoon and representative images of a 3-week co-culture of Cre[+] and reporter[+] B16F1 and B16F10 cell lines, scale bar 100 μm.
J  Quantification of *in vitro* and *in vivo* Cre[+] EV transfer, grand mean of three replicates of three wells (*in vitro*) or three replicate mice, 15 sections each (*in vivo*). *T*-test for *in vitro* co-culture to reporter only and Mann–Whitney for *in vitro*–*in vivo* Cre[+] EV transfer, $n = 3$ independent experiments.

Source data are available online for this figure.

However, the latter one we excluded previously in other tumor models in which cells exchanged Cre activity when located in physically separated tumors (Zomer *et al*, 2015).

Interestingly, whilst EVs are taken up *in vitro* (Fig 1E), in a 3-week co-culture of B16F1-Cre[+] cells and B16F10-reporter[+] cells, and vice versa, we did not observe a substantial number of cells that report Cre activity (< 0.01%; Fig 1I and J). These data suggest that the Cre-Lox system reports the release of cargo into the cytoplasm rather than only the uptake of EVs and that the *in vitro* EV uptake (i.e., uptake of labeled EVs in Fig 1E) did not coincide with substantial functional release of the content (i.e., lack of Cre-mediated color switch in Fig 1I and J). Moreover, the large discrepancy between the efficiency of Cre[+] EV transfer *in vitro* and *in vivo* suggests divergent mechanisms of EV exchange and underlines the importance of studying EV exchange between cells in their *in vivo* setting.

### B16F1 cancer cells have a higher migration speed after uptake of B16F10-derived EVs

Since B16F10 cancer cells have a higher metastatic and migratory capacity than B16F1 cancer cells, we tested whether the migration of B16F1 recipient cells is affected upon the transfer and release of cargo of EVs produced by B16F10 cells. To test this, we considered to study whether inhibition of the release of EVs by B16F10 cells would affect the migratory behavior of B16F1 and B16F10 recipient cells. Unfortunately, good tools to only inhibit EV release without affecting the donor cells do currently not exist. However, as mentioned above, the Cre-LoxP system allows to address exactly this question using an alternative approach: The DsRed[+] cells did not release the luminal EV cargo and will behave similarly to cells that did not receive luminal EV cargo upon inhibition of EV release in the donor cells and can therefore act as a control for cells that do take up EVs (i.e., GFP-expressing cells). To test the effect of EV transfer between the different models, we visualized the migratory behavior of recipient cells by IVM (Fig 2A and B). Tumors consisting of B16F10 Cre[+] cancer cells and B16F1 reporter[+] cancer cells, or B16F1 Cre[+] cancer cells and B16F1 or B16F10 reporter[+] cancer cells were intravitally imaged for 4–6 h. The positions of Cre[+] CFP[+] cells, and the cells that did not (DsRed[+] reporter[+]) or did receive EV cargo (eGFP[+] reporter[+]) were annotated in every image to determine the migration speed (Fig 2A

and B). B16F1 cells that have taken up B16F10 EV cargo (eGFP[+] reporter[+] cells) have a higher migration speed than B16F1 cells that have not taken up this EV cargo (DsRed[+] reporter[+] cells; Fig 2C). By contrast, when the B16F1 or B16F10 cells take up EVs produced by B16F1 cells, the migration speed is not enhanced (Fig 2D and E).

### Isolation of EVs released by cancer cells located in their *in vivo* setting

To identify EV cargo that may explain the phenocopy of the migratory behavior, we isolated EVs from tumors consisting of either B16F1 or B16F10 cells (Fig 3A). Since B16 tumors consist predominately of cancer cells (on average > 70%, Appendix Fig S1), it is expected that the vast majority of EVs in tumors are produced by cancer cells, although we cannot exclude the co-isolation of some EVs derived from non-cancer cells. To isolate EVs, we used a procedure that previously was successfully used for brain tissues (Levy, 2017; Vella *et al*, 2017) based on gentle enzymatic dissociation to release cells and the population of EVs from tumors. Next, we isolated cells and EVs by differential ultracentrifugation (D-UC). We aim to study the total landscape of EVs, instead of focusing just on exosomes, since other EVs such as microvesicles, oncosomes, and apoptotic bodies may also have an important function. Based on the maximum centrifugation speed required to pellet vesicles, we identified two vesicle populations: 16.5K EVs that were pelleted at 16,500 $g$ and 100K EVs that were pelleted at 100,000 $g$ (Fig 3A). Electron microscopy (EM) shows that the 16.5K fraction is enriched for larger EVs (≥ 150 nm, black arrows), and the 100K fraction is enriched for smaller EVs (≤ 150 nm, red arrows; Fig 3B). Both fractions also contain characteristic melanosome structures, known to be released by B16 cells into the extracellular environment (Willms *et al*, 2016). Although melanosomes are generally not considered to be EVs, cancer-associated fibroblasts can get reprogrammed upon uptake of melanosomes released by transformed melanocytes (Dror *et al*, 2016; Garcia-Silva & Peinado, 2016).

To profile the population of tumor microenvironmental EVs, we performed total RNA sequencing and proteome profiling using label-free mass spectrometry. In total, we detected 12,450 and 12,802 transcripts for B16F1 16.5K and 100K EVs, respectively, and 11,696

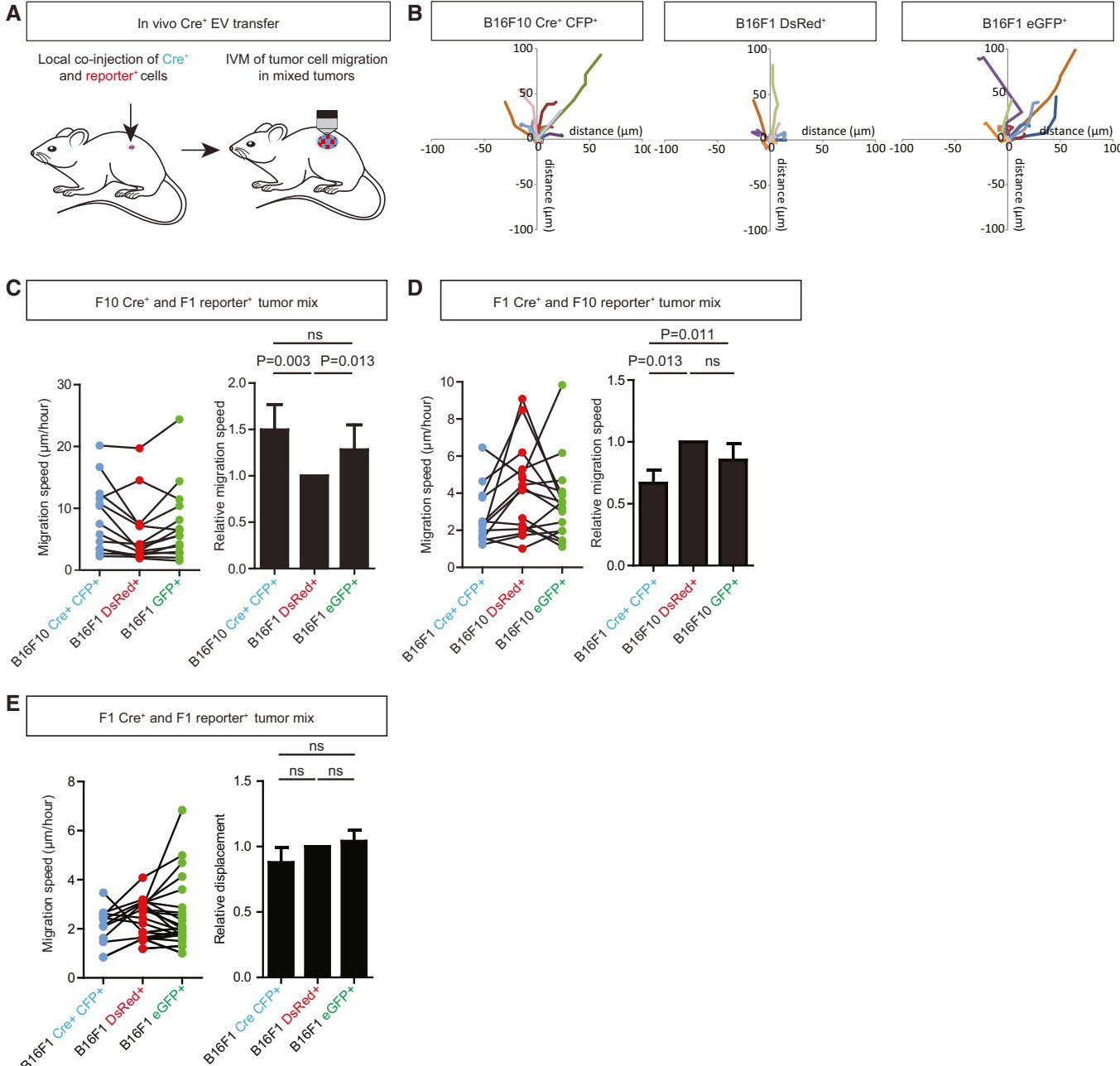

**Figure 2.  B16F1 cells have a higher migration speed after uptake of B16F10-derived EVs.**

A   The effects of *in vivo* EV transfer are studied by local co-injection of Cre[+] and reporter[+] cells and intravital microscopy of established mixed tumors to study tumor cell migration.

B   Representative rose plots of tumor cell migration tracks of B16F10 Cre[+], B16F1 DsRed[+] reporter[+], and eGFP[+] B16F1 reporter[+] tumor cells within the same imaging field, tracks shown for 2h30 migration.

C   Average migration speed of B16F10 Cre[+] and DsRed[+] and eGFP[+] B16F1 reporter[+] cells, *n* = 15 positions in four mice.

D   Average migration speed of B16F1 Cre[+] and DsRed[+] and eGFP[+] B16F10 reporter[+] cells, *n* = 16 positions in four mice.

E   Average migration speed of B16F1 Cre[+] and DsRed[+] and eGFP[+] B16F1 reporter[+] cells, *n* = 22 positions in six mice.

Data information: (C–E) Average speed of cells within the same imaging field is connected with a line (left) and migration speed of cells relative to DsRed[+] reporter[+] migration speed is plotted (right). Data represented as mean ± SEM with the Wilcoxon signed rank test.

and 11,527 transcripts for B16F10 16.5K and 100K EVs, respectively, that were present in all three replicates (Fig EV2A, top and Fig EV3). At the proteome level, we detected 3,210 and 3,333 proteins for B16F1 16.5K and 100K EVs, respectively, and 3,213 and 3,276 proteins for B16F10 16.5K and 100K EVs, respectively, that were present in all three replicates (Fig EV2A bottom and B, and

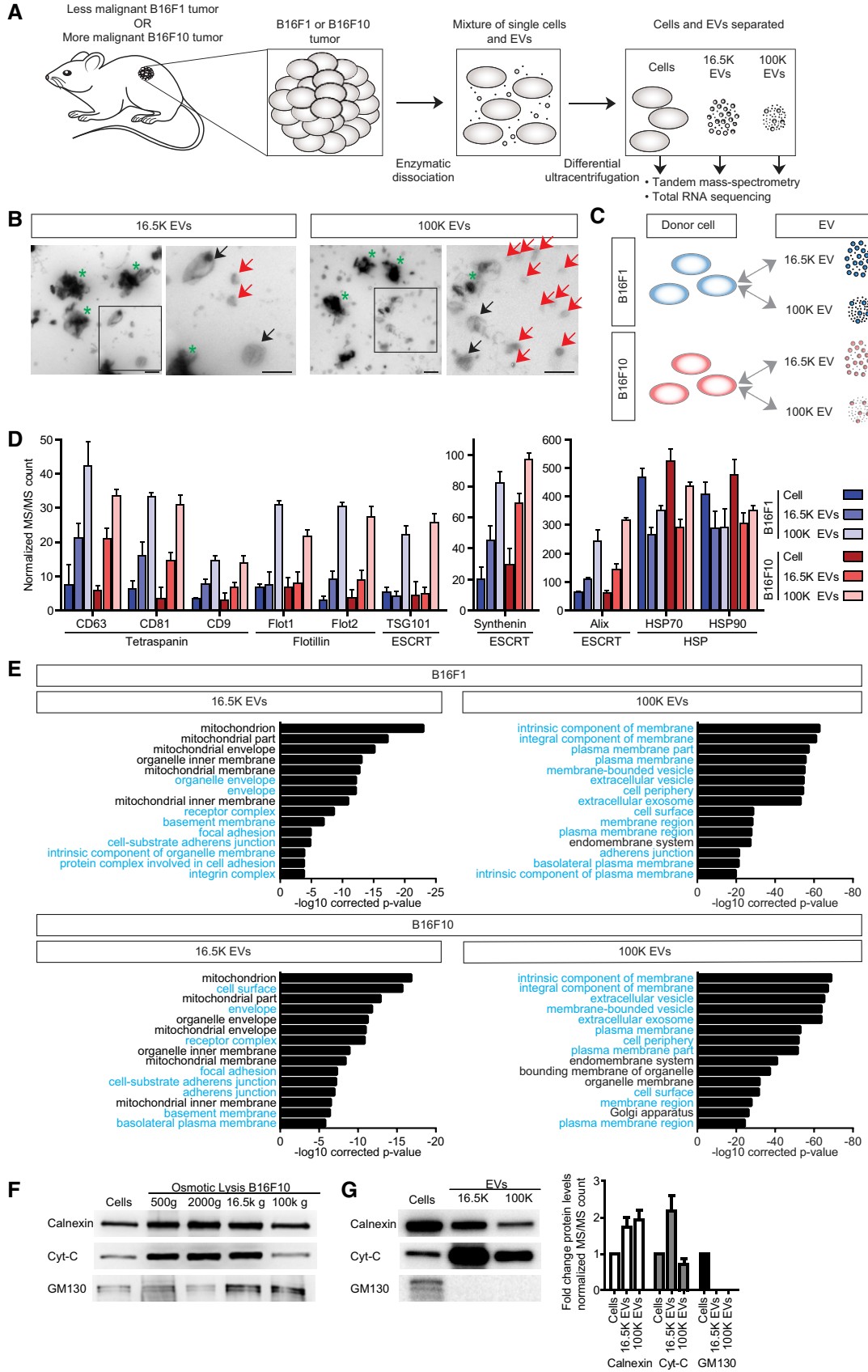

**Figure 3.**

◀

**Figure 3. Successful isolation of EVs from the *in vivo* tumor setting.**

A   Cartoon demonstrating the isolation of cells and EVs directly from tumors grown in syngeneic mice. Tumors are enzymatically dissociated, and cells and EVs are isolated from the single cell and EV mixture by D-UC.

B   Representative electron microscopy images of 16.5K EVs and 100K EVs. Black arrows point to EVs ≥ 150 nm, red arrows point to EVs < 150 nm, and green asterisks mark melanosome-like structures. Scale bar is 500 nm.

C   Cartoon of the sample comparisons used for EV enrichment over the donor cell.

D   Normalized MS/MS count of the classical EV markers tetraspanins, flotillins, ESCRT machinery, and HSP proteins, and data represented as mean ± SD of three independent EV preparations.

E   GO term enrichment for cellular compartment of proteins (FC ≥ 10 and $P ≤ 0.01$) enriched in EVs to the donor cell for 16.5K EVs and 100K EVs of the B16F1 model (top) and B16F10 model (bottom).

F   Western blot of cells and 500 *g*, 2,000 *g*, 16.5K *g*, and 100K *g* fractions of osmotically lysed B16F10 cells for calnexin, cytochrome-C, and GM130. Representative Western blot of three experiments.

G   Western blot (left) and fold change of normalized MS/MS count (right) of cells and 16.5K EVs and 100K EVs for calnexin, cytochrome-C, and GM130. MS/MS fold change represented as mean ± SD of three independent EV preparations.

Data information: Full scans of Western blots from panels (F) and (G) are available as source data.
Source data are available online for this figure.

Fig EV4). To test whether the EVs contained truncated proteins, we analyzed the low molecular weight gel bands of the cell and EV samples that contain small proteins and potentially truncated proteins (gel bands 4 and 5 in Fig EV2B and Appendix Fig S2). This analysis showed that only 3.2% of the 16.5K EV protein cargo and 1.9% of the 100K EV protein cargo are comprised of truncated proteins (Table EV1).

To test whether the isolation of EVs was successful, we investigated the protein cargo of the tumor microenvironmental EVs. A wide variety of EV markers have been published (see ExoCarta and Vesiclepedia), and these markers are often cell type-dependent. Importantly and as expected for EV isolation, we observed enrichment of classical EV markers such as tetraspanins, flotillins, and ESCRT machinery proteins in both EV fractions (Fig 3C and D), with these typical exosome markers showing the highest enrichment in the 100K fraction. Moreover, from the top 100 most often published EV makers described in ExoCarta (Haraszti *et al*, 2016; Keerthikumar *et al*, 2016) and Vesiclepedia (Kalra *et al*, 2012), more than 90% of these EV markers were also present in all our replicates of the 16.5K and 100K EV fractions (Tables EV2 and EV3). Interestingly, we observe that the EVs in different fractions have differential cargo. Proteins shown to have high abundance in DU145 cell-derived 10K EVs (Minciacchi *et al*, 2015) show strong enrichment in our 16.5K EVs but not in our 100K EVs, which may suggest the presence of specific markers per type of EV (e.g., HSPD1, HSPA9, and MDH2, see also Table EV4). A recent study showed a lack of exclusive marker genes for exosomes and microvesicles when EVs from different cell lines were compared (Haraszti *et al*, 2016). In line with our findings, this study found that 100K exosomes were enriched for proteins of receptors and cell adhesion, whereas 10K MVs were enriched in endoplasmic reticulum and mitochondrial proteins (see below). A few other studies have claimed the existence of markers that exclusively differentiate between exosomes and microvesicles (Keerthikumar *et al*, 2015; Minciacchi *et al*, 2015; Kowal *et al*, 2016; See also Table EV5). Our data show that many of these makers are indeed enriched in our 100K EV (e.g., VPS36, ITGA5, and TSG101) or 16.5K EV (e.g., GLS, TUFM, and ETFA) fractions (see also Table EV4). However, these markers are not exclusively present in either one of the fractions, which may be explained by the fact that differential centrifugation does not yield pure fractions.

To study EV proteins in a more unbiased approach, we analyzed the levels of the most prominent proteins enriched in EVs compared to their level in the donor cells ($P ≤ 0.01$ and fold change (FC ≥ 10) using gene ontology (GO) term analysis for cellular compartments). As expected for an EV fraction with small EVs such as exosomes, we observed that 100K EVs are highly enriched for extracellular vesicle and plasma membrane-related GO terms (Fig 3C and E, GO terms highlighted in blue). As expected for larger EVs that bud off from the limiting membrane, thereby capturing a fraction of the cytoplasm including intact organelles (Johnson *et al*, 2017), 16.5K EVs are enriched for envelope, adhesion, and plasma membrane-related GO terms (highlighted in blue), and also mitochondrion (Fig 3C and E). Interestingly, transfer of intact mitochondria between cells has been shown to restore tumorigenic potential in cancer cells, and EVs have been proposed as a mode of mitochondrial transfer (Tan *et al*, 2015; Dong *et al*, 2017). To exclude that classically non-extravesicular organelles, including mitochondria, are present in the 16.5K fraction due to cell shearing during the enzymatic dissociation and D-UC, control cells were purposely lysed by osmotic shock and subjected to the identical D-UC protocol to isolate the 16.5K and 100K fractions. Upon cell lysis, markers for the ER, mitochondria, and Golgi (respectively, calnexin, cytochrome-C, and GM130) are present in all D-UC pellets (Fig 3F). Importantly, in both 16.5K and 100K EV fractions isolated from tumors, the Golgi marker GM130 could not be detected by Western blot and mass spectrometry, illustrating that the contribution of co-isolated organelles from lysed cells in our EV preparations is minor and below the detection level (Fig 3G). All these data together show that we have successfully isolated EVs from solid tumors and that our 16.5K fractions are enriched for large EVs that also contain a fraction of the cytoplasmic content, and our 100K fractions are enriched for smaller EVs such as exosomes.

## Cells and EVs isolated from B16F1 and B16F10 tumors contain distinct sets of RNA and protein

To better understand the molecular differences between the B16F1 and B16F10 melanoma tumors on both the cellular level and EVs, we characterized differential abundance of genes and proteins in cells and in the different EV preparations (Fig EV5A). We found that a similar number of proteins are differentially abundant across the sample types of the B16F1 and B16F10 model (296, 304, and 340 genes across cells, 16.5K EVs, and 100K EVs, respectively; Fig EV5B, right). By contrast, on RNA level we found a higher

number of differentially abundant RNAs in EVs than in cells (65, 105, and 571 RNAs across cells, 16.5K EVs, and 100K EVs, respectively; Fig EV5B, left).

Since we observed differential capacity of migration of B16F1 and B16F10 (Fig 1A–D) and transfer of the migratory phenotype from B16F10 to B16F1 cells (Fig 2C), we analyzed whether EVs carry proteins and RNAs that can influence migratory processes. Gene ontology analysis on EV-RNA and protein showed that the cargo of EVs is associated with many different biological processes including some that can be directly linked to cell migration such as cytoskeleton organization, mesenchymal cell development, mesoderm morphogenesis, and response to axon injury (Fig EV5B). In addition to processes directly linked to migration, we found many processes that can more indirectly influence migratory capacity of a cell, such as regulation of microtubule polymerization, but also to RNA processing and epigenetic and posttranscriptional regulation of gene expression (Fig EV5B). Together, this suggests that EVs could induce a multi-faceted biological response in recipient cells.

To specifically study further differences in cell migration, we performed process-oriented analysis by mapping differentially expressed genes and proteins on the interaction network of cell migration using the String database (Szklarczyk et al, 2017). These interactions are defined as "associations (that) are meant to be specific and meaningful, i.e. proteins jointly contribute to a shared function; this does not necessarily mean they are physically binding each other". These interactions include enzyme/substrate, (transcriptional) regulation, and other indirect interactions between proteins in the same network that influence each other. We only included protein–protein interactions that are experimentally validated and/or annotated in curated databases (Fig 4A and Materials and Methods). Both B16F1 and B16F10 are enriched for a distinguishing set of RNAs and proteins, with more and higher interconnected RNA and protein molecules related to cell migration for the B16F10 model (Fig 4B and C). Notably, the highest enrichment of migration-related RNA and protein is generally observed in EVs, suggesting that EV-mediated transfer of these molecules could result in a concerted action in recipient cells. Together, these data demonstrate that cells with different metastatic potential produce EVs with a cargo that is distinct across the models.

## EVs of the B16F1 and B16F10 models are enriched for RNAs and proteins involved in migration

Interestingly, unsupervised clustering analysis of the cargo showed that both the RNA and protein content of EVs released by B16F1

and B16F10 are more different from the content of the releasing cell than from the content of other EVs (Figs EV3 and EV4), suggesting that the content of the pool of EVs is different from the cellular content of the producing cells. If this is true, the ratios of different biomolecules within cells should differ from the ratio of the same biomolecules in the pool of EVs. To test this, we analyzed the same amount of vesicular and cellular protein and RNA, and subsequently compared the relative abundance of RNAs ($P \leq 0.01$, Log2FC $\geq 1$) and proteins ($P \leq 0.01$, FC $\geq 20$) in EVs and donor cells (Fig 5A). Indeed, we found many RNAs where the abundance in EVs deviates from the abundance in cells, for respectively, B16F1 and B16F10 231 and 249 RNAs in 16.5K EVs, and 1,089 and 1,463 RNAs in 100K EVs (Fig 5B, left). Moreover, this also holds true for proteins, for respectively, B16F1 and B16F10 706 and 632 proteins in 16.5K EVs, and 1,067 and 1,054 proteins in 100K EVs (Fig 5B, right). In both tumor models, the majority of the EV-enriched RNAs and proteins are carried by the 100K EVs, whilst less differentially abundant RNAs and proteins are enriched in the 16.5K EVs. These data are in line with the idea that the 16.5K EVs are enriched for vesicles that bud from the plasma membrane and carry a fraction of the cytoplasm. Therefore, they have less specific RNA and protein loading than the small EVs in the 100K fraction, such as exosomes that are derived from multivesicular bodies (Crescitelli et al, 2013; Witwer et al, 2013; Mateescu et al, 2017).

Because B16F1 and B16F10 cells have differential migration capacity, we analyzed whether cargo that is specifically loaded in EVs can affect migration. GO term analysis showed that the EV-enriched proteins and RNAs are involved in a variety of biological processes (Fig 5). The top GO terms are related to processes known to be upregulated during metastases including cell migration, wound healing, and morphogenesis (Fig 5B, highlighted in red). The highest functional enrichment is observed for cell surface receptor signaling pathway in the 100K EVs (Fig 5B). Previously, other groups have shown that EV-associated receptor tyrosine kinases can be functionally incorporated in the plasma membrane of recipient cells (Al-Nedawi et al, 2008; Zhang et al, 2017). Moreover, > 100 proteins that are linked to cell surface receptor signaling pathways were found to be enriched in EVs (Fig 5B, see also Table EV6), implying that EVs could transfer receptor-mediated oncogenic signaling to other cells. EV enrichment in relation to cell migration and cell surface receptor signaling pathway is similar for EVs released by cells of both models with differential metastatic potential. However, this may not be so surprising since cells of both models, though with a distinct potential, are migratory (Fig 1A–D).

**Figure 4. B16F10 cells and EVs have a higher enrichment of distinct migration-related RNAs and proteins compared to B16F1 cells and EVs.**

A     Schematic flowchart showing how differential expression of migration-related RNAs and proteins between the B16F1 and B16F10 model will be identified. Detected transcripts and proteins were selected for differential expression ($P \leq 0.05$) within at least one sample type (cells, 16.5K EVs, 100K EVs) and only differentially expressed genes or proteins plotted on the interaction network of cell migration, only representing experimentally determined interactions and interactions from curated databases.

B, C   Differential RNA (b) and protein (C) expression of B16F1-specific and B16F10-specific regulators of cell migration. Every node is divided into three partitions for expression in cells (top), 16.5K EVs (right), and 100K EVs (left) and colored for higher expression in B16F1 (blue) or B16F10 (red). Migration-related RNAs and proteins are clustered by annotation as positive regulators (top), negative regulators (bottom), and general cell migration (middle) and arranged for B16F1-specific, B16F10-specific, and shared enrichment across all sample types.

Data information: Edge evidence for (B) and (C) is available as source data.
Source data are available online for this figure.

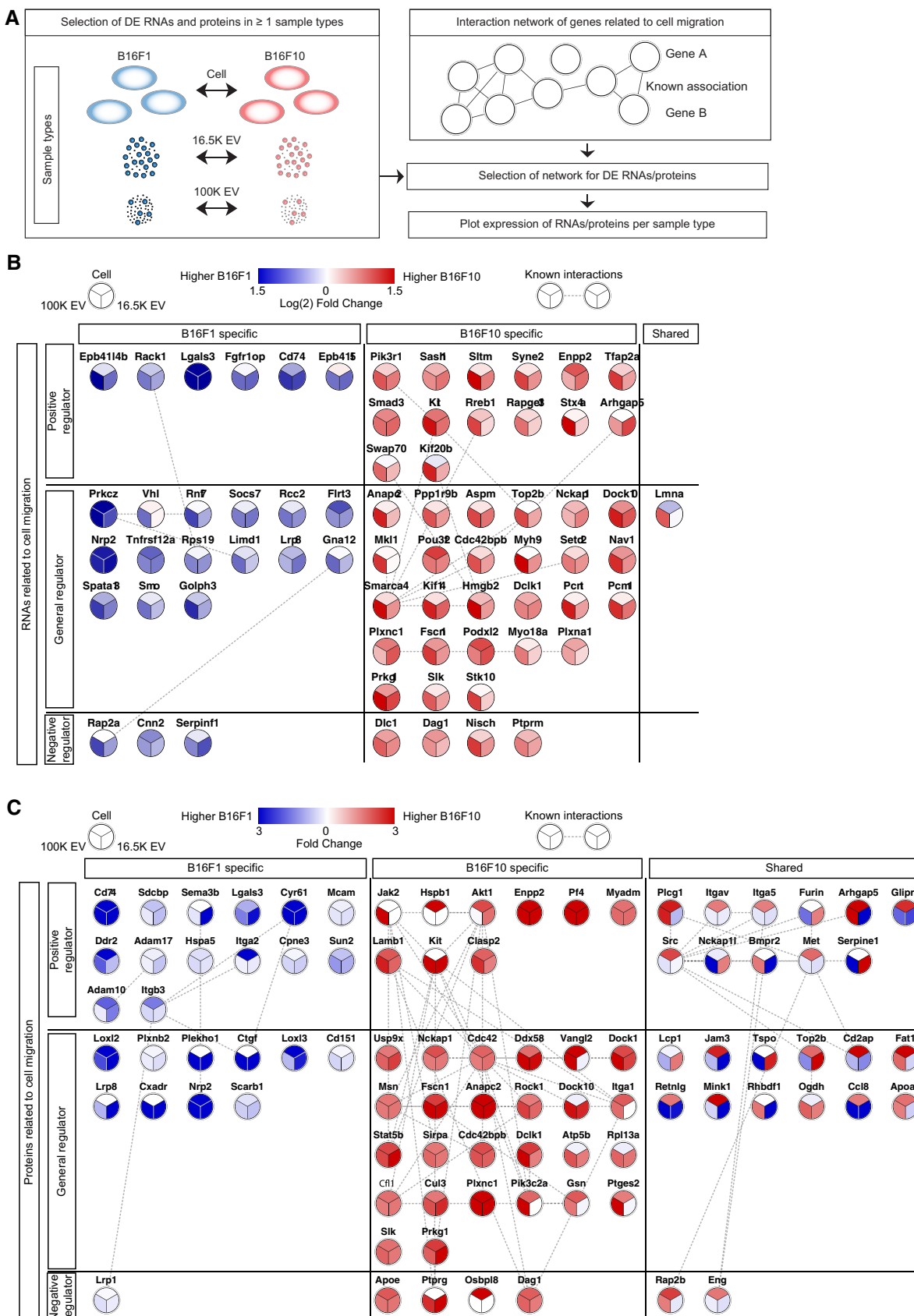

**Figure 4.**

## RNAs and proteins involved in migration have the potential to be functional in recipient cells

Despite specific loading of some of the vesicular RNA and proteins, EV cargo may not all have the same effect in the recipient cells. For example, a biological molecule that is specifically sorted in EVs by a donor cell but is already very abundant in the recipient cell is not likely to add much to the recipient cell. As another example, biological molecules that are passively transferred to EVs can potentially have a large effect if these molecules are at low abundance in the recipient cell. Therefore, solely studying the content of EVs does not show which biomolecules will have a strong effect in the recipient cells. In an attempt to provide theoretical weight for the most effective vesicular RNA and proteins, we identified the RNAs ($P \leq 0.01$, Log2FC $\geq 1$) and proteins ($P \leq 0.01$, FC $\geq 20$) in which the abundance in the EVs deviates most from the abundance in recipient cells (Fig EV6A). Since we found that B16F1 and B16F10 cancer cells have differential migratory capacity, we tested whether the EV cargo with the most theoretical weight can influence migration. GO term analysis showed that hundreds of RNAs and proteins involved in cell surface receptor signaling pathways, migration, wound healing, and morphogenesis have the potential to be more effective upon transfer by EVs (Fig EV6B). To focus on the most relevant EV proteins of this analysis, we integrated the data with the list of differentially expressed EV proteins in the comparison B16F1 versus B16F10 (see below). Moreover, the effectiveness of biomolecules carried by B16F1 or B16F10 EVs does not only depend on the level in the recipient cell, but also on whether the vesicular biomolecules are members of the same signaling pathways and networks so that they can have a concerted action. To test the latter, we plotted the potential effective vesicular RNAs and proteins on the interaction network of cell migration (Fig 6A). To this end, we selected for RNAs ($P \leq 0.05$, FC $\geq 3$) and proteins ($P \leq 0.05$, FC $\geq 5$) enriched in EVs to recipient cells (Fig 6A, arrows marked 1). Next, to focus on differences between the transfer of B16F1 and B16F10 EVs across these different tumor models, we selected genes and proteins for differential enrichment (RNAs FC $\geq 3$, proteins FC $\geq 5$; Fig 6A, arrows marked 2). This shows that theoretically, both B16F1 and B16F10 EVs can transfer a distinct set of RNA and protein across cancer cell subclones (Fig 6B). Most molecules are uniquely transferred from B16F1 EVs to B16F10 cells or B16F10 EVs to B16F1 cells by RNA or protein at these criteria, and some biomolecules can be transferred by both RNA and protein (NRP2, ENPP2, TFAP2a). Rare exceptions to unidirectional transfer are ARSB, NAV1, SERPINE1, and ACVRL1, which can be transferred in both directions, depending on EV subtype or RNA/protein content. Although equal numbers of general cell migration RNA and protein molecules can be potentially effective in the recipient cells, B16F10 EVs theoretically transfer more positive regulators of cell migration,

whereas B16F1 EVs transfer more negative regulators of cell migration to recipient cells (Fig 6B).

Combined, our data confirm that tumor cells with distinct metastatic potential shed EVs containing RNAs and proteins involved in cancer cell migration. Importantly, our results suggest that the effectiveness of the transfer depends on the abundance of these molecules in EVs relative to the recipient cell and on whether these molecules can have a concerted action as a positive function migration network.

## Concluding remarks

Prior studies have focused on the characterization of EVs isolated from biofluids, such as blood, where the vast majority of EVs are released by non-transformed cells, or from the media of *in vitro* cultured cells (Mateescu *et al*, 2017). Our data suggest a distinct efficiency of EV transfer *in vitro* and *in vivo*, and emphasizes the importance of studying the cargo of EVs isolated from the *in vivo* setting. We focused our analysis on both enrichment of small (100K) and enrichment of large (16.5K) EVs. However, it should be realized that these two EV populations may still contain multiple EV types or non-detectable cell fragments that are released upon the isolation procedure (Fig 3F and G). EV subtypes could be potentially further fractionated using density gradient centrifugation (Vella *et al*, 2017) or antibody capture (Kowal *et al*, 2016), yet it remains to be determined what the contribution of these populations is in the *in vivo* tumor microenvironment. In our study, we have focused on the RNA and protein content of EVs and their potential role in the EV-mediated phenocopy of migration behavior. Our methodology did not allow to study the complete EV cargo, and other non-detected active biomolecules may also play a role in the transfer of migratory behavior, such as protein modifications (e.g., phosphorylation), lipids, metabolites, and other RNA species (e.g., miRNA and lncRNA).

Importantly, analysis of the EV cargo demonstrated the presence of typical EV markers. By isolating EVs present in the *in vivo* tumor setting and global molecular characterization of the cargo, we show that cancer subclones with distinct metastatic behavior release pools of EVs that transfer several nodes (RNAs and proteins) of networks involved in migration. Importantly, these cancer subclones have a shared origin and the same pre-existing signaling networks. Therefore, EV cargo does not "transplant" completely new networks into the recipient cells, but instead amplifies existing nodes in the network already present in the cell. In addition to networks of migration-related RNA and protein, we also found that EVs carry biomolecules with diverse other functions. For instance, EVs of the B16F10 model are enriched for epigenetic and posttranscriptional regulation of gene expression, as well as (m)RNA processing, potentially bringing about diverse effects in cells that receive these EVs (Fig EV5B). Moreover, RNA molecules related to glycolysis (Figs 5B

**Figure 5.  EVs of B16F1 and B16F10 tumors are enriched for a specific set of transcripts and proteins compared to the donor cells.**

A   Cartoon of the comparison of EVs to the respective donor cell.
B   Differential expression in EVs to the respective donor cell in the B16F1 model (top) and B16F10 model (bottom) for most discriminating gene expression levels (Log2FC $\geq 1$, $P \leq 0.05$, left) and proteins (FC $\geq 20$, $P \leq 0.01$ right). For every comparison, number of DE genes and proteins, supervised clustering of samples, and gene ontology for functional enrichment in EVs are depicted. GO terms related to cell migration are highlighted in red, and indirect influencers of cell migration are highlighted in green.

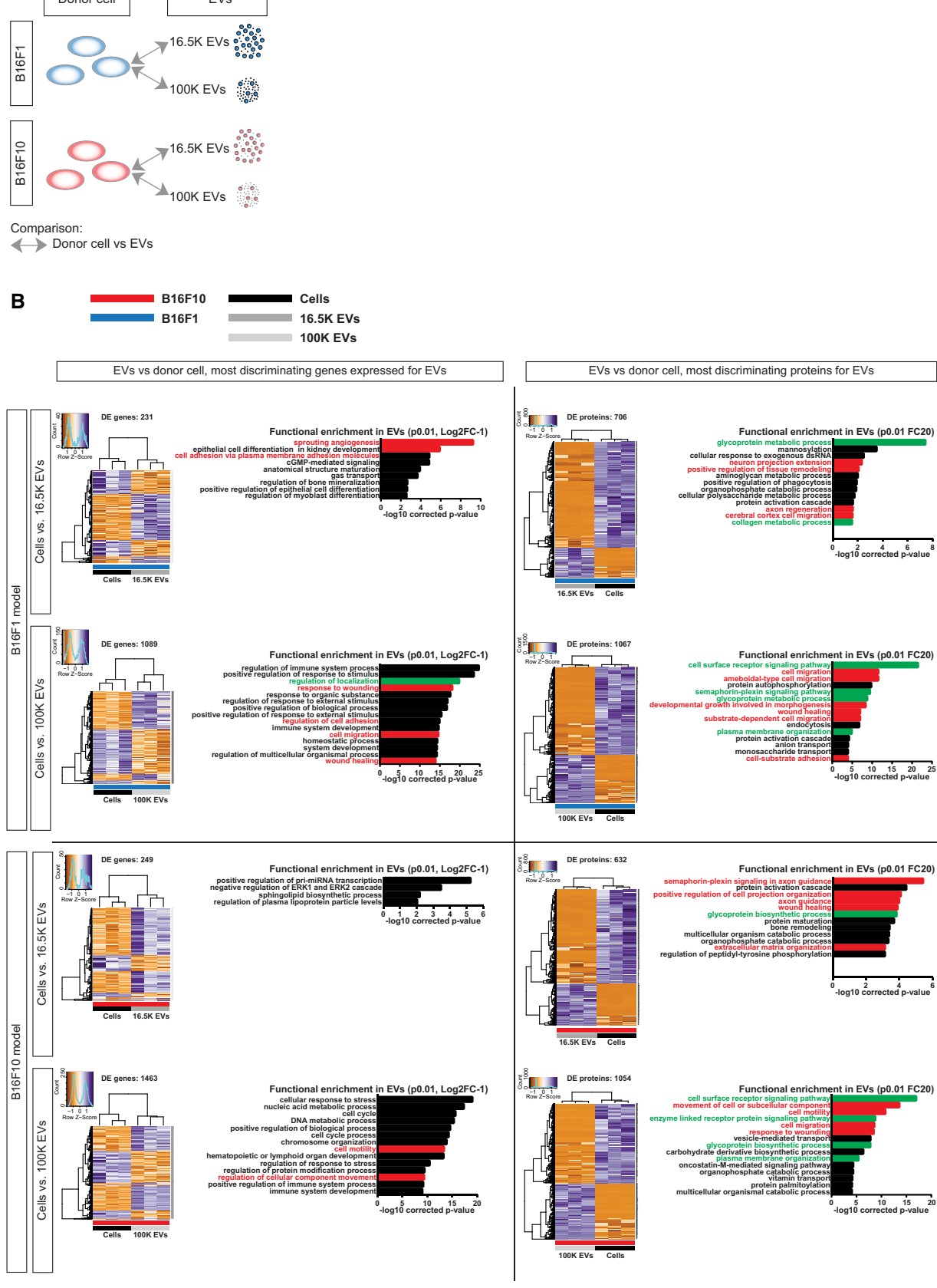

**Figure 5.**

and EV6B; glycoprotein metabolic process) are enriched in EVs. Interestingly, glycolysis has been shown to be upregulated in migratory cancer cells (Han *et al*, 2013; Shiraishi *et al*, 2015). Additionally, enrichment of receptor-mediated signaling in EVs (cell surface signaling pathway, Figs 5B and EV6B, and Table EV6) may help to transfer oncogenic properties between cancer cells. Together, the transfer of molecules involved in the above-mentioned processes may further amplify the effectiveness of the transferred biomolecules involved in migration, concertedly invoking a multi-faceted biological response. Importantly, the balance of positive and negative regulators in individual EVs and the balance of this cargo in all EVs taken up by a recipient cell determine what, if any, effect EV uptake has on the recipient cell (Gho & Lee, 2017). For instance, if the pool of EVs has as many positive regulators as it has negative regulators of equal strength, the uptake of these EVs on these processes would potentially be neutral. As single EVs cannot contain all cell-expressed RNAs and proteins due to volume restrictions (Sverdlov, 2012), future studies will have to identify whether and how the total EV cargo is divided over individual EVs of the same and different subtypes. In order to become motile in the primary tumor, cancer cells upregulate multiple parallel networks of signaling pathways involved in migration (Fig 4B and C; Wang *et al*, 2004, 2005; Wyckoff *et al*, 2007). Importantly, analysis of migratory cells isolated from tumors with different genetic origins showed that upregulation of different genes in the same pathway can be altered in different cells to achieve the same migratory behavior (Condeelis *et al*, 2005; Wyckoff *et al*, 2007). Our data show that the pool of EVs released by tumor cells transfers networks of interconnected RNAs and proteins. These data illustrate that it may not be the upregulation of the level of one particular RNA or protein that mediates the phenotypic change in recipient cells, but it could be the upregulation of the activity of migratory pathways as a whole. The consequence of this finding is that the output in the recipient cells will be very robust; even if the recipient cells do not receive all biomolecules or only a small amount of these molecules as can be transported by EVs, the concerted action of the network will lead to the same migratory output of the recipient cell. Only if the whole loading of EVs is changed, for example due to a changing expression of hundreds to thousands of genes in the donor cell upon a gene alteration (Verdoni *et al*, 2008; Eraly, 2014; Ding *et al*, 2015; Gerstung *et al*, 2015), the response of the recipient cell will change. Therefore, the nature of networks and concerted action of EV-associated cargo, where interchangeability of biomolecules gives the same output, could render the pool of tumor EVs a robust intracellular messenger that controls local and distant cell behavior.

# Materials and Methods

## Tissue culture

B16F1 (obtained from ATCC) and B16F10 (a kind gift of Prof. Tom Würdinger) cells were cultured in full medium: DMEM, high glucose, and GlutaMAX (Life Technologies, cat. no. 31966-021) supplemented with 10% FBS (Sigma-Aldrich, cat. no. F7524) and 50 U/ml penicillin and 50 μg/ml streptomycin (Penicillin-streptomycin; 5,000 U/ml; Life Technologies, cat. no. 15070-063). Cre[+] and reporter[+] cells were made as previously described (Zomer *et al*, 2016). In short, B16F1 or B16F10 cells were transfected with plasmid pcDNA3.1-CFP; Cre25nt; Zeo (Addgene, plasmid number 65727) using Lipofectamine 2000 (Life Technologies, cat. no. 11668-019) according to the manufacturer's protocol. One day after transfection, 125 μg/ml Zeocin (100 mg/ml, Invitrogen, cat. no. 46-0509) was added to select cells expressing the construct, followed by selection of CFP[+] cells using FACS. Monoclonal CFP[+] cell lines were stained for Cre with a mouse anti-Cre antibody (Millipore, cat. no. MAB3120) to select cell lines with a high expression of the Cre-recombinase. Reporter[+] cells were made by lentiviral transductions with pLV-CMV-LoxP-DsRed-LoxP-eGFP (Addgene, plasmid number 65726), and selection with 5 μg/ml puromycin (Life Technologies, cat. no. A11138-03), followed by selection of DsRed[+]/eGFP[−] cells using FACS to obtain reporter[+] cells that do not display eGFP[+] background recombination.

## *In vitro* co-culture assays

B16F1 and B16F10 cells were plated using a 10:1 ratio of Cre[+]:reporter[+] cells. Co-cultures were maintained for 3 weeks and split twice a week to prevent co-culture overgrowth. After 3 weeks, fluorescent images of the co-cultures were obtained with a Leica AF7000 microscope for CFP (excitation 430/24, emission 470/40), GFP (excitation 470/40, emission 520/40), and DsRed (excitation 572/35, emission 640/50). Percentages of eGFP[+] reporter[+] cells were determined as described before (Zomer *et al*, 2016).

## Uptake of PKH-stained EVs

For *in vitro*-derived EV isolation, B16F1 or B16F10 cells were cultured in full medium as described above but supplemented with 10% EV-depleted FBS (16 h 100,000 *g* centrifugation) instead of 10% regular FBS. At 90% confluence, conditioned medium was collected and EVs isolated using differential ultracentrifugation (D-UC). Conditioned medium was centrifuged 2× at 500 *g* for 10 min

**Figure 6. EVs from B16F1 and B16F10 cells transfer distinctive migration-related RNA and protein molecules to putative recipient cells.**

A   To identify EV cargo that can elicit a functional response in recipient cells, EV content was filtered on enrichment over putative recipient cells (RNA: $P \leq 0.05$, FC $\geq 3$; protein: $P \leq 0.05$, FC $\geq 5$, arrow indicated by 1). Next, to filter out EV cargo enriched to cells independent of tumor model, cargo was selected for differential EV enrichment in B16F1 EV to B16F10 cell and B16F10 EV to B16F1 cell (RNA: $\geq 3$-fold higher directional enrichment, protein: $\geq 5$-fold higher directional enrichment, arrow indicated by 2).

B   Interaction network of cell migration-related proteins and genes differentially transferred between B16F1 and B16F10 tumor cells through EVs. Migration-related genes and proteins are clustered by annotation as positive regulators, negative regulators, and general cell migration. Split nodes representing RNA (yellow edge) or protein (green edge) display if the biomolecule is transferred by 16.5K EVs (left), 100K EVs (right), or both EV populations in gray. Experimentally determined interactions and interactions from curated databases (see also source data for this figure) are represented by dotted lines between nodes. RNA or protein with shared transfer between 16.5K EVs and 100K EVs or with shared transfer between RNA and protein is marked by an asterisk (Arsb, Nav1, Serpine1, Acvrl1).

Source data are available online for this figure.

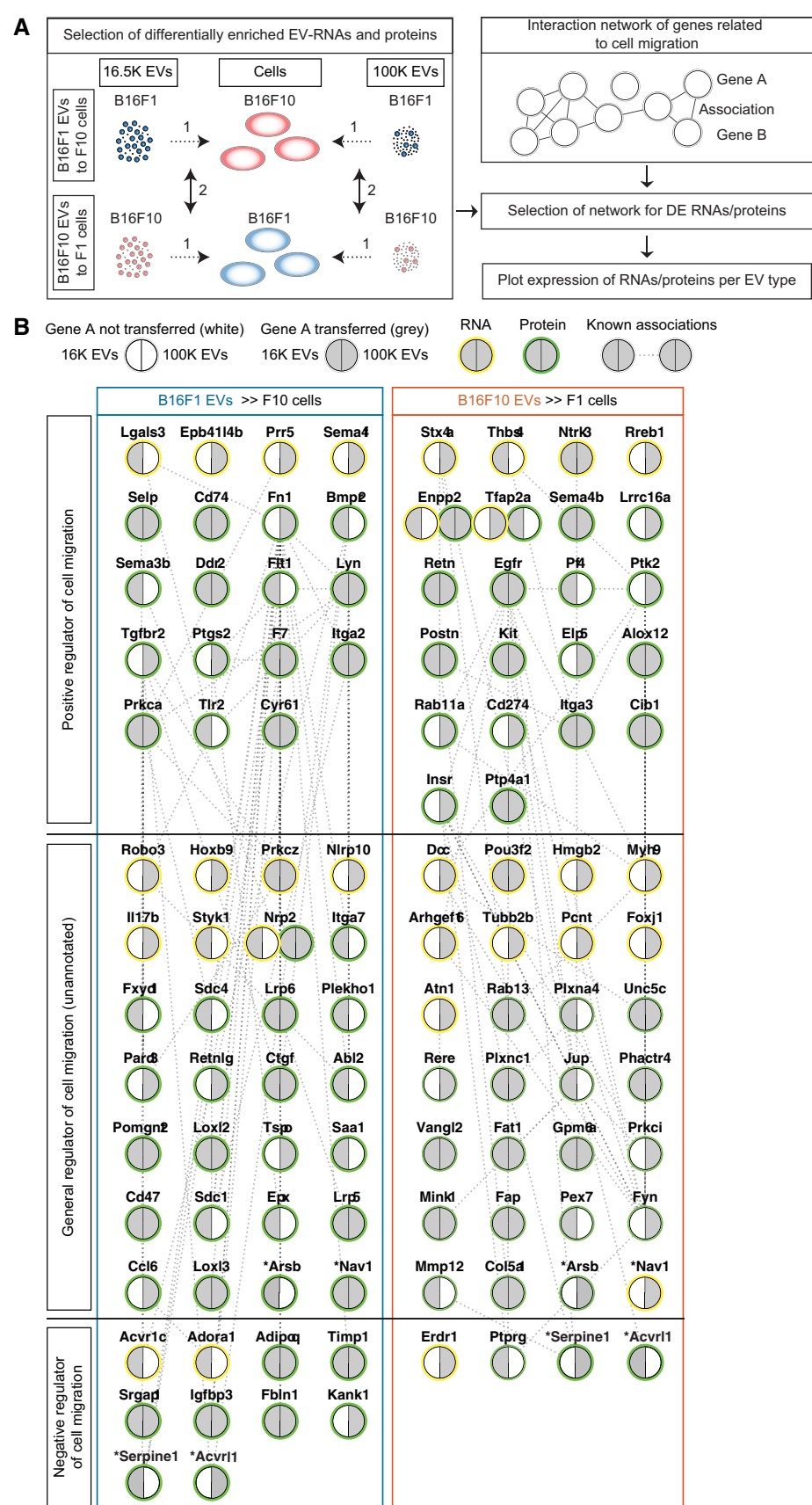

**Figure 6.**

and 2× at 2,000 *g* for 15 min to remove cells and cell debris. 16.5K EVs were pelleted by 16,500 *g* for 24 min, supernatant was cleared by extra centrifugation at 16,500 *g,* and 100K EVs were pelleted by 100,000 *g* centrifugation for 70 min, subjected to a PBS wash followed by final 100,000 *g* centrifugation for 70 min. 16,500 *g* and 100,000 *g* steps were carried out in an Optima L-90K ultracentrifuge (Beckman Coulter) in combination with a SW 32 Ti rotor (Beckman Coulter) and Ultra-Clear tubes (Beckman Coulter, cat. no. 344058).

PKH staining of EVs was performed with the PKH staining kit (Sigma-Aldrich, MINI67-1KT) by taking up the 16.5K or 100K EV pellet in 100 µl Diluent C and addition of 100 µl Diluent C with 1.5 µl PKH-67, followed by thorough mixing. After 5 min, 37 ml of EV-depleted full medium was added to the PKH-stained EVs, and EVs were re-pelleted by centrifugation at 16,500 *g* for 24 min or 100,000 *g* centrifugation for 70 min. PKH-stained EVs were added to B16F1 reporter[+] or B16F10 reporter[+] cells in eight-well µ-slides (Ibidi cat. no. 80827) as indicated, and medium was replaced with full medium after 3 h. Internalized EVs were visualized on a Leica SP5 confocal microscope equipped with a 63× glycerol N.A. 1.30 objective at 488 nm excitation for PKH and 561 nm excitation for DsRed. Internalization of PKH-stained EVs was confirmed by making z-stacks from cell bottom to cell top with a step size of 250 nm.

## Mice

Black 6 (B6) mice (own crossing) were housed under standard housing conditions and received *ad libitum* food and water. Experiments were carried out in accordance with the guidelines of, and approved by, the Animal Welfare Committee of the Royal Netherlands Academy of Arts and Sciences, the Netherlands.

## *In vivo* tumor mixes and intravital microscopy

B16F1 and B16F10 cells in PBS were injected subcutaneously or intradermally in the back of B6 mice of 8–20 weeks old. A total of 100,000 cells were injected per site at a ratio of 2:1 to 10:1 for B16F10Cre[+]:B16F1reporter[+] and at a ratio of 2:1 to 20:1 for B16F1Cre[+]:B16F10reporter[+]. For reporter[+] only negative controls, a total of 100,000 reporter[+] cells were injected. To image developed tumors, mice were anesthetized using medical oxygen with 1–2% of isoflurane and tumors were exposed by a surgical skin flap procedure as previously described (Zomer *et al*, 2015). Images were acquired using a Leica SP5 microscope with 405 or 458 nm excitation for CFP, 488 nm excitation for eGFP, and 561 nm excitation for DsRed. To study cancer cell migration, 4- to 6-h time-lapse imaging was acquired with a 30-min interval and corrected for XYZ-drift using custom software. Cells were tracked using manual tracking in FIJI. Migration tracks and migration speed were calculated in Excel.

## Quantification of percentage of cancer cells within the tumor and *in vivo* EV transfer

Tumors were isolated from mice implanted with B16F10 Cre[+] and B16F1 reporter[+] tumors as described above. Tumors were fixed using a periodate–lysine–PFA buffer and frozen in OCT (Leica, cat. no. 14020108926) as described before (Zomer *et al*, 2016).

For quantification of percentage of cancer cells within the tumor, 10-µm-thick sections were stained with 0.1 µg/ml DAPI in PBS and

imaged using a Leica SP5 microscope with 405 nm excitation for DAPI and 561 nm excitation for DsRed. Using TissueQuest software (TissueGnostics), the DAPI channel was used for nuclear segmentation. Nuclear regions were assessed for expression of DsRed in a −1 and +1 µm ring mask around the nuclear edge, and all nuclei were scored positive (cancer cell nucleus) or scored negative (stromal cell nucleus) for DsRed.

For quantification of *in vivo* EV transfer, 100-µm-thick tumor sections were cut and imaged using a Leica SP5 microscope with 405 or 458 nm excitation for CFP, 488 nm excitation for eGFP, and 561 nm excitation for DsRed. Of the tumors, the ratio of eGFP/DsRed was determined by thresholding in ImageJ as described before (Zomer *et al*, 2016).

## *In vivo* EV isolation

100,000 B16F1 or B16F10 cells in PBS were injected subcutaneously into 8- to 20-week-old B6 mice. Before mice reached the humane endpoint, mice were sacrificed and tumors isolated. Tumors were chopped in 3-mm cubes, washed in PBS, and centrifuged at 500 *g* for 4 min. Tumor pieces were digested into a single-cell suspension by incubation in with 25 µg/ml DNase I (Roche, cat. no. 10104159001) and 5 Wünsch units of Liberase (Roche, cat. no. 05 401 151 001) in PBS at 37°C for 20 min whilst shaking. The single-cell suspension was strained with a 70 µm strainer whilst washing with 5 mM EDTA (Life Technologies, cat. no. 15576-028) and 25 µg/ml DNase I in PBS. Cells were pelleted by centrifugation at 500 *g* for 10 min. Supernatant was subjected to an additional centrifugation of 500 *g* for 10 min and 2× at 2,000 *g* for 15 min to remove residual cells and cell debris. 16.5K EVs were pelleted by 16,500 *g* for 24 min, and EVs were washed with PBS followed by final pelleting at 16,500 *g*. Supernatant was cleared by an extra centrifugation at 16,500 *g,* and 100K EVs were pelleted by 100,000 *g* centrifugation for 70 min subjected to a PBS wash followed by final pelleting at 100,000 *g* centrifugation for 70 min. Cell and EV samples were split for analysis by EM, Western blot, mass spectrometry, and RNA sequencing.

For later EM and Western blot analyses, EVs were stored in PBS at −80°C. For later mass spectrometry, samples were directly taken up in 1× NuPAGE LDS sample buffer (Novex NP0007) with 10% 1 M DTT (Promega 0000085706) and boiled for 5 min at 100°C before storage at −80°C. For RNA processing, samples were directly mixed with 800 µl Trizol Reagent (Life Technologies 15596018), mixed well, and incubated at RT for 5 min before storage at −80°C.

## Reverse transcriptase PCR

Cell and EV samples from the tumor microenvironment were collected as described above. RNA isolation was performed on cell and EVs previously stored in Trizol at −80°C. RNA isolation was performed according to the manufacturer's instructions with addition of 5 µl glycogen to the aqueous phase (Roche, 10901393001). RNA pellet was taken up in RNase-free water. cDNA was prepared using the High-Capacity cDNA Reverse Transcription kit (Applied Biosystems 4368814) according to the manufacturer's instructions. cDNA was amplified using primers for Cre (Forward primer 5′ GCCTGCATTACCGGTCGATGC 3′; Reverse primer 5′ GTGGCAGATG GCGCGGCAACA 3′) and RPL38 (Forward primer 5′ AGGATGC

CAAGTCTGTCAAGA 3′; reverse primer 5′ TCCTTGTTGTGATAAC CAGGG 3′) using the thermal cycles 5 min at 95°C, 35 cycles of 95°C for 30 s, 58°C for 30 s, 72°C for 1 min, and after the last cycle a final extension of 10 min at 72°C. PCR products were visualized on a 2% TAE agarose gel.

## Osmotic lysis

For osmotic lysis, cells were washed with MilliQ and lysed by a 20-min incubation in MilliQ. Cell lysis was confirmed by phase-contrast microscopy, and PBS was added to the lysed cells in MilliQ. Lysed cells were subjected to the identical D-UC protocol as described under "*In vivo* EV isolation", in short 2× 500 *g*, 2× 2,000 *g*, 2× 16,500 *g*, and 2× 100,000 *g*. Samples were collected at the first centrifugation step of respective isolation speed and stored in PBS at −80°C until further processing for Western blot.

## Transmission electron microscopy (TEM)

Transmission electron microscopy (TEM) imaging of EVs was carried out as previously described (Kooijmans *et al*, 2016). Briefly, EVs were diluted in PBS and absorbed to carbon-coated formvar grids (Cell Microscopy Center (CMC), Utrecht, the Netherlands). Fixation of EVs was achieved with 0.2% glutaraldehyde and 2% paraformaldehyde in 100 mM phosphate buffer (pH 7.4). Grids were negatively stained with uranyl oxalate and embedded in methyl cellulose–uranyl acetate. Images of EVs were obtained by using a Tecnai T12 electron microscope (FEI, Eindhoven, the Netherlands).

## Western blot

Cells and EVs were lysed in 1% SDS and 10 mM EDTA lysis buffer at 100°C for 5 min. Samples were subsequently boiled at 100°C for 5 min in 1× NuPAGE LDS sample buffer (Novex NP0007) with 10% 1M DTT (Promega 0000085706), and 20 µg of protein per sample was loaded onto a 4–15% Mini-PROTEAN TGX gel (Bio-Rad 4561083). Proteins were transferred to a PVDF membrane using wet transfer, and the membrane was blocked using 5% milk. Membranes were probed for calnexin (Abcam ab22595), cyto-chrome-C (BD Biosciences 556433), and GM130 (BD Biosciences 610822), incubated with anti-mouse HRP (GE Healthcare NA931V) or anti-rabbit HRP (GE Healthcare NA934V), and proteins were detected using ECL Western Blotting Substrate (Pierce 32209) in combination with an ImageQuant LAS 4000 digital imager (GE Healthcare). As a reliable loading control for cells and EVs has not yet been identified, equal protein loading was confirmed using Ponceau S staining and lanes were quantified using ImageJ.

## Sample preparation for LC-MS/MS

Protein lysates (50 µg) were separated on precast 4–12% gradient gels using the NuPAGE SDS-PAGE system (Invitrogen, Carlsbad, CA). Following electrophoresis, gels were fixed in 50% ethanol/3% phosphoric acid solution and stained with Coomassie R-250. Gel lanes were cut into five bands, and each band was cut into ~1 mm³ cubes. Gel cubes were washed with 50 mM ammonium bicarbonate/50% acetonitrile and were transferred to a 1.5 ml microcentrifuge tube, vortexed in 400 µl 50 mM ammonium bicarbonate for

10 min, and pelleted. The supernatant was removed, and the gel cubes were vortexed in 400 µl 50 mM ammonium bicarbonate/50% acetonitrile for 10 min. After pelleting and removal of the supernatant, this wash step was repeated. Subsequently, gel cubes were reduced in 50 mM ammonium bicarbonate supplemented with 10 mM DTT at 56°C for 1 h, the supernatant was removed, and gel cubes were alkylated in 50 mM ammonium bicarbonate supplemented with 50 mM iodoacetamide for 45 min at room temperature in the dark. Next, gel cubes were washed with 50 mM ammonium bicarbonate/50% acetonitrile dried in a vacuum centrifuge at 50°C for 10 min and covered with trypsin solution (6.25 ng/µl in 50 mM ammonium bicarbonate). Following rehydration with trypsin solution and removal of excess trypsin, gel cubes were covered with 50 mM ammonium bicarbonate and incubated overnight at 25°C. Peptides were extracted from the gel cubes with 100 µl of 1% formic acid (once) and 100 µl of 5% formic acid/50% acetonitrile (twice). All extracts were pooled and stored at −20°C until use. Prior to LC-MS, the extracts were concentrated in a vacuum centrifuge at 50°C, and volumes were adjusted to 50 µl by adding 0.05% formic acid, filtered through a 0.45 um spin filter, and transferred to an LC autosampler vial.

## LC-MS/MS

Peptides were separated by an Ultimate 3000 nanoLC-MS/MS system (Dionex LC-Packings, Amsterdam, the Netherlands) equipped with a 20 cm × 75 µm ID fused silica column custom packed with 1.9 µm 120 Å ReproSil Pur C18 aqua (Dr Maisch GMBH, Ammerbuch-Entringen, Germany). After injection, peptides were trapped at 6 µl/min on a 10 mm × 100 µm ID trap column packed with 5 µm 120 Å ReproSil Pur C18 aqua in 0.05% formic acid. Peptides were separated at 300 nl/min in a 10–40% gradient (buffer A: 0.5% acetic acid (Fisher Scientific), buffer B: 80% ACN, 0.5% acetic acid) in 60 min (90-min inject-to-inject). Eluting peptides were ionized at a potential of +2 kVa into a Q Exactive mass spectrometer (Thermo Fisher, Bremen, Germany). Intact masses were measured at resolution 70,000 (at *m/z* 200) in the orbitrap using an AGC target value of 3E6 charges. The top 10 peptide signals (charge-states 2+ and higher) were submitted to MS/MS in the HCD (higher-energy collision) cell (1.6 amu isolation width, 25% normalized collision energy). MS/MS spectra were acquired at resolution 17,500 (at *m/z* 200) in the orbitrap using an AGC target value of 1E6 charges, a maxIT of 60 ms, and an underfill ratio of 0.1%. Dynamic exclusion was applied with a repeat count of 1 and an exclusion time of 30 s.

## Protein identification

MS/MS spectra were searched against the Uniprot Mus musculus reference proteome FASTA file (release June 2015, 42,296 entries, canonical and isoforms, no fragments) supplemented with the Cre-recombinase sequence using MaxQuant 1.5.2.8 (Cox & Mann, 2008). Enzyme specificity was set to trypsin, and up to two missed cleavages were allowed. Cysteine carboxamidomethylation (Cys, +57.021464 Da) was treated as fixed modification and methionine oxidation (Met, +15.994915 Da) and N-terminal acetylation (N-terminal, +42.010565 Da) as variable modifications. Peptide precursor ions were searched with a maximum mass deviation of 4.5 ppm

                                                                                      

and fragment ions with a maximum mass deviation of 20 ppm. Peptide and protein identifications were filtered at an FDR of 1% using the decoy database strategy. The minimal peptide length was 7 amino acids. Proteins that could not be differentiated based on MS/MS spectra alone were grouped into protein groups (default MaxQuant settings). Searches were performed with the label-free quantification option selected.

## Label-free quantitation

Proteins were quantified by spectral counting, i.e., the number of identified MS/MS spectra for a given protein (Liu *et al*, 2004). Raw counts were normalized on the sum of spectral counts for all identified proteins in a particular sample, relative to the average sample sum determined with all samples. To find statistically significant differences in normalized counts between sample groups, we applied the beta-binomial test (Pham *et al*, 2010), which takes into account within-sample and between-sample variation using an alpha level of 0.05.

## Identification of degradation products in EVs

For each sample type (cell lysate, 16.5K EVs, 100K EVs), protein data were exported for each individual gel block: fractions 1–1.5, 1.5–2.5, 2.5–3.5, 3.5–4.5, and 4.5–5. For each gel block, the predicted molecular weight of all identified proteins was plotted. Next, to identify outliers in the lower molecular weight blocks, proteins were identified that have a predicted molecular weight $\geq 2\times$ the standard deviation of gel blocks 4 and 5 (fraction 3.5–5), which resulted in a cutoff of 71.35 kDa. Utilizing these criteria, outlier proteins were identified in cell lysate (22), 16.5K EVs (104:3.2% of 16.5K EV-identified proteins), and 100K EVs (63:1.9% of 100K EV-identified proteins, with a portion of the outlier proteins identified as outliers in EVs and not in the cell lysate [16.5K EVs: 98 (3.1% of 16.5K EV-identified proteins), 100K EVs: 59 (1.8% of 100K EV-identified proteins)].

## Transcriptomics

RNA isolation was performed on cells and EVs previously stored in Trizol at −80°C. RNA isolation was performed according to the manufacturer's instructions with addition of 5 μl glycogen to the aqueous phase (Roche, 10901393001). RNA pellet was taken up in RNase-free water. Samples were submitted to the Utrecht sequencing facility for Truseq RNA stranded ribo-zero library prep and sequencing on the Illumina NextSeq500 1 × 75 bp High Output (300M) platform. After sequencing quality control, mapping and counting analyses were performed using our in-house RNA analysis pipeline v2.1.0 (https://github.com/UMCUGenetics/RNASeq), based on best practices guidelines (https://software.broadinstitute.org/gatk/guide/article?id=3891). In short, sequence reads were checked for quality by FastQC (v0.11.4) after which reads were aligned to GRCm38 using STAR (v2.4.2a) and add read groups using Picard perform quality control on generated BAM files using Picard (v1.141). Samples passing QC were then processed to count reads in features using HTSeq-count (v0.6.1). After read counting (ENSEMBL definitions GRCm38, release 70), genes with low expression (mean count < 5) were removed (26,295 out of 38,293), resulting in a

11,998 gene by 18 sample count matrix with ENSEMBL gene identifiers. The median-of-ratios method from the DESeq2 R package (Love *et al*, 2014) was used to normalize all samples for sequencing depth. ENSEMBL identifiers were mapped to gene symbols using the biomaRt R package (Durinck *et al*, 2009). Unsupervised clustering was performed on normalized read counts (blinded dispersion estimation, variance stabilizing transformation) using Euclidean distance and complete hierarchical clustering using the heatmap.2 package.

## GO term analysis

GO term analysis for cellular compartment or biological process on most differentiating proteins and genes was performed with Cytoscape 3.3.0 using the ClueGO 2.2.4 plugin with default settings. To reduce redundancy in cellular compartment GO terms, GO term fusion was applied and the top 15 groups plotted with the corrected group *P*-value. To reduce redundancy in biological process GO terms, GO term fusion and GO term grouping were applied and the top 15 groups plotted with the corrected group *P*-value. For RNA EV enrichment to donor or recipient cells of the other model, most discriminating genes were selected by Log2FC $\geq 1$ and $P \leq 0.01$. For RNA enrichment between B16F1 and B16F10 within a sample type (i.e., cells–cells, 16.5K EVs-16.5K EVs, and 100K EVs-100K EVs), genes were selected by Log2FC $\geq 1$ and $P \leq 0.05$. For protein EV enrichment to donor or hypothetical recipient cells, most discriminating proteins were selected by FC $\geq 10$ and $P \leq 0.01$. For protein enrichment between B16F1 and B16F10 within a sample type (i.e., cells–cells, 16.5K EVs-16.5K EVs, and 100K EVs-100K EVs), proteins were selected by FC $\geq 1.5$ and $P \leq 0.05$.

## Interaction networks

Proteins associated with biological processes were obtained from the Cytoscape plugin ClueGO 2.2.4 in Cytoscape 3.3.0. Associated proteins were loaded into String (Szklarczyk *et al*, 2017), networks exported as text format, unconnected nodes added back to the network, and networks imported into Cytoscape 2.8.3. Complete networks were filtered on proteins detected by mass spectrometry and/or RNA sequencing. Only edges representing evidence of known interactions from curated databases (Biocarta, BioCyc, Gene Ontology, KEGG, and Reactome) and experimentally determined interactions (DIP, BioGRID, HPRD, IntAct, MINT, and PDB) were preserved with protein homology to other species. For differential expression between sample types, networks were further selected on differential expression between the B16F1 and B16F10 model in $\geq 1$ sample category with adjusted *P*-value $\leq 0.05$. Expression was plotted in blue (higher B16F1) or red (higher B16F10) for sample comparisons cell, 16.5K EVs, and 100K EVs using the Cytoscape plugin MultiColoredNodes 2.5.40. Nodes were organized in B16F1-specific, B16F10-specific, or shared categories as appropriate and interactions removed if strings spanned across expression categories. To identify EV cargo that can elicit a functional response in recipient cells, EV content was filtered for enrichment to cells of the other model with FC $\geq 3$ (RNA) or FC $\geq 5$ (protein) and $P \leq 0.05$. Next, differential enrichment was filtered for a $\geq 3$-fold (RNA) and $\geq 5$-fold (protein) directional enrichment. Fulfillment of these criteria was plotted in gray for 16.5K and 100K EVs using the

Cytoscape plugin MultiColoredNodes 2.5.40. Nodes were organized in B16F1 EV to B16F10 cell transfer and B16F10 EV to B16F1 transfer.

### Statistical analysis

Statistical analysis was performed using GraphPad Prism 5. Data were tested for normal distribution using the D'Agostino and Pearson omnibus normality test. For normally distributed data, the unpaired *t*-test was used. For not normally distributed data, the Mann–Whitney test was performed. For paired measurements with not normally distributed data, the Wilcoxon matched pair test was used.

### Data availability

The mass spectrometry proteomics data have been deposited to the ProteomeXchange Consortium via the PRIDE partner repository with the dataset identifier PXD006439. Additionally, peptide and protein data are available in Datasets EV1 and EV2, respectively. The total RNA-seq data have been deposited to the European Nucleotide Archive (ENA) with the project identifier PRJEB20729 and can be directly accessed through http://www.ebi.ac.uk/ena/data/view/PRJEB20729.

**Expanded View** for this article is available online.

### Acknowledgements

We thank all members of the van Rheenen group for their input in experiments and this manuscript, and we thank Anko the Graaff and the Hubrecht Imaging Center for imaging support. We also thank the Utrecht Sequencing facility for the total RNA sequencing, and Annelies Barendregt-Smouter for assistance in analysis of transcriptomic data. This work was supported by the European Research Council (consolidator grant Cancer-Recurrence 648804 and InCeM 642866 to JvR, and CoMMiTMenT 602121 to CRJ), research grants from the Dutch Organization of Scientific Research (NWO; 823.02.017), Cancer Genomics Netherlands, the Doctor Josef Steiner Foundation (to J.v.R), and the European Union's Horizon 2020 Research and Innovation Program Under the Marie Sklodowska-Curie grant agreement No 642866.

### Author contributions

JvR, with the help of CRJ, conceived and supervised the study. JvR, CRJ, AZ, and SCS designed the experiments. SCS performed most of the experiments. AZ made the cell lines and assisted with the IVM and cell migration analysis. RH and RMS performed electron microscopy. JdL and EC performed quality control and gene expression analysis of transcriptomic data. TVP, JCK, SRP, TS, and CRJ performed LC-MS/MS, protein identification, and label-free quantitation. JvR, SCS, CRJ, SRP, and JdL wrote the manuscript, which was reviewed and approved by all authors.

### Conflict of interest

The authors declare that they have no conflict of interest.

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
