## [Review Process File · The EMBO Journal]

Cancer cells copy migratory behavior and exchange signaling networks via extracellular vesicles

Sander C. Steenbeek, Thang V. Pham, Joep de Ligt, Anoeck Zomer, Jaco C. Knol, Sander R. Piersma, Tim Schelfhorst, Rick Huisjes, Raymond M. Schiffelers, Edwin Cuppen, Connie R. Jimenez, Jacco van Rheenen

Review timeline:

Submission date:	5 October 2017
Editorial Decision:	18 December 2017
Revision received:	29 January 2018
Editorial Decision:	21 March 2018
Revision received:	2 May 2018
Accepted:	17 May 2018

Editor: Daniel Klimmeck

Transaction Report:

1st Editorial Decision

18 December 2017

Thank you for the submission of your manuscript (EMBOJ-2017-98357) to The EMBO Journal, as well as giving additional input in your preliminary point-by-point response. I apologise for the delay in getting back to you due to detailed discussions in the team. As mentioned, your study has been sent to three referees, and we have received reports from all of them, which I copy below.

The referees acknowledge the potential interest and novelty of your work, although they also express major concerns. While referee #2 is overall more positive, referee #1 states that your claims on functionally interdependent RNA-protein networks and exchange of cargo-loaded exosomes are not sufficiently well supported by the data. Further, this referee asks you to broaden and consolidate the bioinformatic analysis of your data and validate your findings with functional experiments. Referee #3 agrees in that the study lacks validation in his/her view and that the transfer of concerted networks is not demonstrated. In addition, the referees point to a number of inconsistencies between data and missing controls, which would need to be resolved to achieve the level of robustness needed for The EMBO Journal.

I judge the comments of the referees to be generally reasonable and can - based on your sensible preliminary point-by-point response - offer to invite you to revise your manuscript experimentally to address the referees' comments. We would need strong support from the referees on such a revised version of the manuscript to move towards publication.

Please note however, that while the points related to RNA-protein network interactions and proof of exosomal transfer of the newly identified cargo (referee #1, pts. 1, 6; ref#3 pt. 1) are per se well taken, taking into account the scope of the current work and character as a resource article, we consider these requested experiment as not essential for this manuscript.

In more detail, we ask you to

- carefully address the concerns of referee #1 and #3 regarding network character, functional causalities and exosomal origin and rewrite the manuscript accordingly pointing to potential limitations of the data.
- put your work better into the literature context, eg regarding transferred RNA translation-stability and destination (Ref #1, pt.1), as well as proof of Cre-reported EV transfer per se (Ref #3, pt. 2).
- clarify which claims are directly derived from the current study, and tone down the ones supported by literature.
- Complement and generalize the bioinformatic analysis of the RNAseq-proteome data (Ref #3, standfirst and pts. 3,4,8; ref#3, pts. 3,4), and integrate the conclusions of the resource part with the in vivo migration assay.
- Increase mouse numbers to improve robustness of in vivo metastatic potential data (Figure S6; ref#3, pt.5) or tone down claims on this aspect.

Referee #1:

In this study, Steenbeek et al investigate the reciprocal communication between cells that differ in metastatic potential, but present within the same tumor environment. As a model system, they use the syngenic cell lines B16F1 and B16F10, and determine RNA and protein cargo in extracellular vesicles (EVs) shed by either cell type. The authors find remarkable complexity and differences in EV RNA and protein make-up, dependent on cellular origin and type of EV considered (obtained by differential centrifugation). This is further interpreted modeling these data as functional networks, mainly in categories related to cell motility, resulting in protein and mRNA networks. The authors conclude by showing that EVs originating from B16F10 cells induce enhanced migration by uptake of EVs originating from B16F1 cells.

This study addresses an important question, and technically it is well-performed (in vivo migration assays, proteomics). They identify an amazingly large number of proteins and transcripts in EVs, with an equally large functional diversity. These are important, especially in conjunction with the compositional difference of 16.5k and 100k EVs. However my main concern is a conceptual one, where the authors propose a model for which in the end they do not provide any direct evidence. Specifically, the concept that the authors are aiming to get across (and put in the title) is that the EV-content should be seen as a network, and not merely as a collection of proteins/mRNAs. This may be appealing as a model, however the authors do not really substantiate this beyond drawing connections between proteins/RNAs that co-occur within a certain gene-ontology/functional category. First of all, this is not surprising since this is how GO-terms are defined. Second, a network is defined by interdependencies, meaning that if the network is perturbed it loses functionality. Proving this will involve careful and not necessarily straightforward experimentation, however none of this is attempted and therefore the network-theory remains a theory: it is not conclusive, and there are no data in the manuscript that cannot be explained by a simpler model, namely that the trans-effect is mediated simply by the collection of molecules contained within EVs.

Other remarks:

1. The authors do not discuss how they envision mRNA to transmit function (if any) in recipient cells. Are RNAs stable throughout the process of EV transfer, including internalization? Are they translated?
2. Figure 1E: it is surprising that EVs remain intact after uptake. Have the authors investigated their ultimate destination, and where/when they dissipate (if at all)?
3. Figure 3: the authors restrict themselves by considering proteins/transcripts involved in cell migration. However, the number of proteins in EVs is so high, and their functionality so diverse, that multiple other cellular processes are ignored that could be equally important to effectuate a trans-effect to recipient cells. So if the authors want to propose EVs to represent networks, Fig 3 (and 5) represent only a small portion of such a network.
4. Figure 3B: what do these 'associations' mean, how are they defined? Direct interactions, enzyme/substrate, (transcriptional) regulation, co-occurring in the literature? Authors need to clarify this - for instance bottom of page 9: 'the highest enrichment of migration-related RNA and protein is

generally observed in EVs, suggesting that EV-mediated transfer of these molecules could result in a concerted action in recipient cells". This really depends what the 'association' between these genes/proteins indicate. If it is transcriptional regulation of one by the other, transfer of both proteins does not have a meaning in the sense of a concerted action.

5. Related to the previous point, authors state (page 9-10): 'The highest functional enrichment is observed for cell surface receptor signaling pathway in the 100K EVs, implying that EVs could transfer receptor-mediated oncogenic signaling to other cells'. This is somewhat of a simplistic view, since RTKs only function if they are properly positioned in the membrane, to sense growth factors and induce a response via second messengers etc. I.e. they only function when the whole machinery is in place, for which no evidence is provided.

6. Page 11-12: Here the authors seem to perform an in-silico analysis of what proteins/RNAs are transferred between B16F1 and B16F10. It is unclear what can be concluded without experimentally demonstrated that the mentioned molecules are indeed transferred, and have a function.

7. Fig 6CD: have the observed effects been corrected for the number of EVs that have been taken up per cell?

8. Page 14: 'tumor cells transfer networks of interconnected RNAs and proteins'. What does 'interconnected' mean? This relates to my main concern: is it proteins in the same pathway, or the RNAs that encode them? This only makes sense when authors assume that an entire pathway or other functional module is 'transplanted' into a recipient cell. This really cannot be concluded from the motility experiments (Fig 6).

9. More generally speaking, there is a disconnect between proteome/sequencing data and functional analysis. Strictly speaking, Fig 6 could also be figure 1 since none of the acquired knowledge was necessary or helpful to do the cell migration experiment. Even more, it remains unclear to which EV-type the observed effect can be ascribed (16.5k, 100k or something else), leave alone to which protein/RNA they contain.

10. The final conclusion is very sketchy where authors talk about possible consequences of changes in EV-cargo due to mutations or otherwise, however this remains very speculative without having data or tools to observe and quantify these effects.

Referee #2:

The article by Steenbek et al provides a very thorough comparative study of the protein and mRNA composition of EVs of different biophysical characteristics, isolated in vivo from a more and a less metastatic subclone of a given tumor cell line. Refined bioinformatics analyses of these extensive array of data leads to the hypothesis that EVs can transfer clusters of RNA and proteins acting in the same physiological pathways, to promote e.g. migration of less migratory cells. The authors validate this hypothesis by showing enhanced in vivo migration of cells that have captured pro-migratory EVs.

This article follows on a very elegant previous study of this group (Zomer, Cell 2015), where the authors had implemented use of the Cre/LoxP system to evidence functional transfer of Cre mRNA between two human tumor cell lines. The experimental system used here is more relevant to the clinical situation, as it uses two subclones of a tumor cell line, rather than two completely different tumor cell lines, thus mimicking the intrinsic tumor heterogeneity of a tumor. It is also implemented in fully immunocompetent mice, instead of relying on injection of human tumor cells in nude mice. The article provides a large range of very valuable information for the EV and cancer fields, especially the extensive proteomic and transcriptomic analyses of EV subtypes from different sources. It confirms and extends recent studies from other groups highlighting the functional relevance of different subtypes of EVs, not only "exosomes"! It also provides an important observation that Cre transfer is not efficient in vitro, and only observed in vivo (figure 1). It is certainly of high value as a resource article

I would suggest an additional experiment, that I think is missing in the current manuscript to properly interpret the migration data upon EV transfer: quantification of Cre-mediated recombination (as in figure 1H) and resulting migration (as in Fig 6) in reporter cells co-injected with Cre-donor cells of the same subclone, ie B16F1-Cre + B16F1-reporter. Indeed, the observed increased migration of B16F1-reporter upon capture of EVs from B16F10-donors is significant but not major, and B16F1-derived EVs also contain a majority of pro-migratory networks (Fig5B), even if slightly more anti-migratory networks than B16F10-derived EVs. Thus, can the authors demonstrate that capture of these F1 EVs does not induce as much or as statistically significant

migration in F1 cells as transfer of F10-EVs, which would support their model of EV-mediated phenocopy, or not (as much migration in both cases), in which case the phenocopy model does not hold. However, even if the latter result is observed, the major importance and value of the article as resource remains.

I also have minor comments, that could be taken into account to improve the manuscript.

1) Characterisation of the B16F10 and F1 models and their growth *in vivo*, mentioned in the beginning of results, should be shown as supplementary data

2) Figure 1i should be displayed as dot plots and not histograms, to show distribution of the eGFP+ cell number in individually analyzed tumors/mice.

3) I did not understand exactly the analysis displayed in figure 5: I suppose that the enrichment on EVs over "putative recipient cells" (figure legend) was searched for in the proteomic and transcriptomic data generated for the rest of the article, and not on novel experiments where, for instance, proteomic/transcriptomic analysis would have been done on cells that had captured or not EVs and thus recombined or not, but I am not entirely sure. Can the authors try to explain better in the text their reasoning in this part of the analysis? For instance, in p11, the sentence "we selected for RNA and proteins enriched in EVs to recipient cells" is not very meaningful.

4) the proteomic data comparing EVs of different natures (16,5g vs 100,000 g) could have been valuably discussed in view of other recent studies comparing also different EV subtypes: what parts of the authors' results are consistent with these previous studies (eg Keerthikumar, *Oncotarget* 2015; Minciocchi *Oncotarget* 2015; Willms *Sci Rep* 2016; Kowal *PNAS* 2016; Haraszti *JEV* 2016)? Of note, Lamp2 is presented as a "typical exosome marker" "specifically enriched in the 100k fraction" (text p8), which does not match figure 2D (Lamp2 is more abundant in 16,5 than 100K) and the literature (Lamp2 is not consistently found associated to EVs, it is even described as excluded from exosomes in some studies).

5) the authors should check carefully all figures, for possible mistakes in symbols or presentation: example in fig3c, where panel "positive regulator/B16F1 specific" shows pie-charts of Adam10 and Itgb3 displayed upside-down (Cell at bottom), and/or maybe left-right inversion? In Figure 4B, panel "Cells vs 100K, B16F1", cells are on the right and 100K on the left, whereas in all other panels of this comparison "EVs vs donor cell, .. gene..." cells are positioned on the left. Also scales of "-log10 corrected p-values" are -2... -8 or -5... -20 in some graphs, and +5... +20 or +2... +10 in others: is it normal?

6) although nice and providing interesting resource data, the protocol used to isolate EVs from *in vivo* grown tumors is not as refined as that used by Vella, *JEV* 2017, as it does not involve any further separation of EV subtypes in a density gradient. Hence, the EVs recovered are mixed and could also include some cell fragments, the authors should acknowledge that.

7) It would be really interesting to determine (in a future work) if the negative and positive networks of cell migration found in the B16F1 EVs are in fact present in different EV subtypes, which, if differentially captured by target cells, would then efficiently induce either a pro-migratory, or an anti-migratory effect, rather than the suggested mixed induction of contradictory functions.

Referee #3:

Steenbeek et al. have studied extracellular vesicles (EVs) as regulators of cancer cell migratory behavior. Their manuscript contains a very detailed characterization of the contents of the EVs. In particular, these EVs are isolated from tumors derived from two sister cell lines - a metastatic and a non-metastatic cell line - and not, as is otherwise often the case, from cells in culture. Thus, this characterization has merit. However, the information gained from the analysis of the EV contents is entirely descriptive. The functional data that is included in the paper is not connected to the characterization of the EVs. Furthermore, the functional data don't add anything substantially new to the functional data previously published by the same group (Zomer et al., 2015). There is no validation convincingly showing that any of reported "omics-based" findings has functional consequence. Regrettably, the EV field doesn't really have the tools to effectively test the premise of

the paper, namely that transfer of a network of molecules in EVs - from one cancer cell population to another - is a means for transfer of the ability to metastasis between cancer cell populations.

Additional major issues:

1) The biggest problem of the manuscript is that there is no attempt to prevent EV transfer, or transfer of a specific network of molecules between the cell lines. One would need to show that interfering with EV mediated transfer alters the phenotype of migration to make any conclusion on the importance of the transfer of molecules by EVs. Such experiment are of course very difficult to perform. In addition, the in vitro phenotype of transfer of Cre is rather subtle, so the phenotypic read-out is also difficult to work with. Additionally, although the authors show transfer of EV associated membrane labeling dyes, there is no evidence that transfer of the proteins and RNA that they identify as part of EV actually occurs, not even in vitro.

2) The manuscript contains no convincing evidence that EVs transfer proteins or RNA between cancer cell populations in vivo. The Cre-lox based tracking system is clever, but since there is no data documenting that Cre protein or mRNA is packed into EVs - and only can be transferred between cells through EVs - the method is not conclusive proof that transfer of molecules between cell lines occur through EVs. It is concerning that when the Cre-donor cell line is cultured with the reporter line in vitro then there is almost no transfer of Cre - even though release and uptake of EVs is convincingly shown under these conditions. Also concerning is the more efficient transfer of Cre between the cell lines in vivo. That could be because EV transfer is more efficient in vivo, as suggested by the authors, but there is no data to support this and Cre might just as well be transferred by other means than EVs. In addition, there are no controls to show that there is no spontaneous color switch in the reporter cells in culture or when these are injected into animals without the Cre+ cells. Figure 6 shows that transfer of Cre affects the migration of the B16F1 cells, but it doesn't show that Cre is transferred by EV, or that the altered migration has anything to do with transfer of a network of molecules. In addition, there are no controls with measurement of migration of DsRed+ B16F1 cells or B16F10 cells in "non-co-injected" tumors.

3) The analysis of the contents of the EVs comes across as forced, almost biased, towards showing a connection to migration, including using things like the "reciprocal ratio of abundance". It is not obvious (e.g., in Fig. 3) that regulators of migration are specifically enriched in the EVs. It just looks like there are general differences. What statistical evidence is there that EVs contain regulators of migration - as opposed to other pathways?

4) Are there really 1000 different proteins and 12,000 RNA species packed into EVs? These numbers suggest that the selection of content of EVs is very unspecific. Is it even possible that one EV particle can contain that many different proteins and RNAs? Or do these numbers reflect that different EV particles contain only a subset of the proteins and RNAs. If so, how does that affect the idea of transfer of a network of regulators? With the proteomics data in hand, the authors appear to have an excellent data source to determine whether EVs mostly carry peptide degradation products or full length proteins. This is a major question in the field - and degradation products could have very interesting functions irrespective of the function of the native proteins.

5) The results of the analysis of the effects of co-injection of B16F1 and B16F10 cell lines in the one mouse that developed metastasis from the B16F1 cells are over-stated - when only one mouse out of 11 develops metastasis after co-injection, there is no statistical support for the idea that co-injection is important and the analysis of the resulting metastasis impossible to interpret. In addition, an effect of co-injection wouldn't have to be through transfer of EVs. Finally, no data is presented on metastasis in mice only injected with B16F1 cells.

6) It is great that the authors isolated EVs from tumors and not just the tumor cell lines grown in vitro - but because of that, they cannot conclude that the characterized EVs originate from the cancer cells. They might very well also come from stromal cells. This point isn't even discussed.

1st Revision - authors' response

29 January 2018

Comments from the editor:

• *carefully address the concerns of referee #1 and #3 regarding network character, functional causalities and exosomal origin and rewrite the manuscript accordingly pointing to potential limitations of the data.*

Network character:

First, we have better clarified what interaction means (see below), and second, what the evidence is for these interactions (see below).

First the definition of interactions. The interactions used in the original manuscript represent protein-protein associations exported from the quality-controlled STRING database (Szklarczyk et al., Nucleic Acids Res., 2017). This analysis was performed online at the following website: string-db.org. In this database (STRING), interactions are defined as: “associations (that) are meant to be specific and meaningful, i.e. proteins jointly contribute to a shared function; this does not necessarily mean they are physically binding each other”. The reviewer is correct that this also includes enzyme/substrate, (transcriptional) regulation, and other indirect interactions between proteins in the same network that influence each other. In the revised manuscript, we now define the interaction at page 12: *“These interactions are defined as “associations (that) are meant to be specific and meaningful, i.e. proteins jointly contribute to a shared function; this does not necessarily mean they are physically binding each other”. These interactions includes enzyme/substrate, (transcriptional) regulation, and other indirect interactions between proteins in the same network that influence each other.”*

Second, the evidence for the interactions: in the original manuscript, these interactions from STRING also included predicted interactions such as gene neighborhood, gene fusions, gene co-occurrence, and other interactions based on text-mining, co-expression and protein homology. We now realized, and agree with the reviewer, that not all interactions have the same importance (for example, importance of experimentally determined is higher than importance of text-mined interactions). Therefore, in the revised manuscript we used the STRING database to only select for interactions that are based on experimental evidence. To this end, we have re-analyzed the evidence for the interactions and have updated Figure 3B, C and Figure 5B to only represent edges supported by evidence from experimentally curated databases (Biocarta, BioCyc, Gene Ontology, KEGG and Reactome) and experimentally determined interaction databases (DIP, BioGRID, HPRD, IntAct, MINT, and PDB).

This new analysis confirms that most interactions originally identified are experimental and/or known interactions, confirming that the identified biomolecules in Figure 3 and Figure 5 can act as a molecular machinery. To further support our conclusion that we have indeed identified a migration signaling network, we have also included the raw edge evidence in the supplemental “SourceDataForFigure3” and “SourceDataForFigure5”.

To make the network evidence of the revised figures clear to the reader, in the results and conclusion section on page 12 we have added: *“We only included protein-protein interactions that are experimentally validated and/or annotated in curated databases (Figure 3A and Materials and Methods)”*

Additionally, we mention the EDGE evidence in the figure legends of Figure 3 and 5: *“differential expressed genes or proteins plotted on the interaction network of cell migration, only representing experimentally determined interactions and interactions from curated databases”*. Moreover, we have updated the material and methods section on page 29 with: *“Only edges representing evidence of known interactions from curated databases (Biocarta, BioCyc, Gene Ontology, KEGG and Reactome) and experimentally determined interactions (DIP, BioGRID, HPRD, IntAct, MINT, and PDB) were preserved with protein homology to other species”*

Functional causalities:

A biological molecule loaded in EVs is not per definition important if it is specifically sorted into EVs. For example, a biological molecule that is specifically sorted in EVs but also very abundant in

the recipient cell is not likely to add much to the recipient cell (as an analogy: an extra glass of water in the sea would not have an effect). By contrast, other vesicular biomolecules that may not be specifically sorted into EVs can likely have a large effect if this molecule is at low abundance in the recipient cell. Therefore, solely studying the content of EVs does not show which biomolecules will have a strong effect in the recipient cells. Accordingly, it is required to take into account the abundance of each biomolecule in the recipient cell, and that is what we did in Figure 5. We are strongly convinced that this analysis helps the reader to understand how to interpret the importance of each vesicular biomolecule for inducing a phenotypic change in the recipient cell. We do agree with the reviewers that this analysis only provides a theoretical weight to each biomolecule that can never be experimentally proven, and in the revised manuscript we have emphasized this more.

Extracellular vesicle origin:

We expect that the vast majority of EVs in tumors are produced by cancer cells. First, from cancer cell line and patient data, it is known that cancer cells release more EVs than non-transformed cells and second, cancer cells form the vast majority of cells in B16F1 and B16F10 tumors. In the revised manuscript, we stress this point at page 7 and 8: *“Although B16 tumors consist of both non-cancer cells and cancer cells that produce EVs, it is expected that the vast majority of EVs in tumors are produced by cancer cells. First, from cancer cell line and patient data, it is known that cancer cells release more EVs than non-transformed cells (Cesi, Walbrecht et al., 2016, Logozzi, De Mito et al., 2009),. Acidic pH (Parolini, Federici et al., 2009) and hypoxia (King, Michael et al., 2012), both common characteristics of solid tumors, are thought to contribute to this increased EV release. Second, dysregulation of exocytic pathways and changes in plasma membrane organization have also been suggested to enhance EV release by cancer cells (Minciacchi, Freeman et al., 2015a). Third, cancer cells also release large EVs with a diameter of 1-10 μm , often referred to as oncosomes (Di Vizio, Morello et al., 2012, Minciacchi, You et al., 2015b). Lastly, cancer cells form the vast majority of cells in B16F1 and B16F10 tumors, and therefore the vast majority of EVs present in B16 tumors are derived from cancer cells.”*

Nevertheless, we never intended to imply that we exclude co-isolation of stromal EVs and we have mentioned this in the revised manuscript. In addition, our EV isolation protocol does not involve any further separation of EV subtypes by e.g. a density gradient. Therefore, we stress on page 16 of the revised manuscript that it should be realized that our two EV populations may still contain multiple EV types or non-detectable cell fragments that are released upon isolation procedure; *“(…) it should be realized that these two EV populations may still contain multiple EV types or non-detectable cell fragments that are released upon the isolation procedure (Figure 2F, G). EV subtypes could be potentially further fractionated using density gradient centrifugation (Vella et al., 2017) or antibody capture (Kowal et al., 2016)”*

• *put your work better into the literature context, eg regarding transferred RNA translation-stability and destination (Ref #1, pt.1), as well as proof of Cre-reported EV transfer per se (Ref #3, pt. 2).*

In the introduction, we have expanded on the current state of the literature: *“EV-associated biomolecules such as EV-RNA are stable in EVs and functional upon delivery into recipient cells. For example upon EV uptake, vesicular mRNA is translated into functional proteins (Valadi, Ekstrom et al., 2007), and vesicular miRNAs suppresses target genes in recipient cells (Fong, Zhou et al., 2015, Hergenreider, Heydt et al., 2012, Zhang, Zhang et al., 2015). Moreover, EV RNA-based reporter systems have confirmed the transport of functional mRNA for Cre (Ridder, Sevko et al., 2015, Zomer, Maynard et al., 2015, Zomer, Steenbeek et al., 2016) or GlucB (Lai, Kim et al., 2015) into recipient cells.”*

In addition, we made clear to the readers of our manuscript that in previous papers, we have performed all the controls to confirm EV-Cre transfer and to exclude the transfer of non-EV

associated Cre. In the revised manuscript, we made this clear by adding: “(...) *the DsRed⁺ cells serve as an internal control for cells that have not taken up EVs, providing a similar control to inferring with EV mediated transfer by e.g. using vesiculation-deficient EV donor cells. We have previously shown that Cre is transferred between cells independent of cell-cell fusion and cell-cell contacts, and that Cre RNA is loaded into EVs (Zomer et al., 2015, Zomer et al., 2016). Moreover, reporter cells report the uptake of Cre (DsRed to eGFP switch) upon exposure to purified Cre⁺ EVs, but not upon exposure to Cre protein or lysed Cre⁺ cells (Zomer et al., 2015, Zomer et al., 2016)*”. Moreover, we also added: “*we considered to study whether inhibition of the release of EVs by B16F10 cells would affect the migratory behavior of B16F1 and B16F10 recipient cells. Unfortunately, good tools to only inhibit EV release without affecting the donor cells do currently not exist. However, as mentioned above, the Cre-LoxP system allows to address exactly this question using an alternative approach: the DsRed⁺ cells did not take up EVs and will behave similarly to cells that did not take up EVs upon inhibition of EV release in the donor cells, and can therefore act as a control for cells that do take up EVs (i.e. GFP expressing cells)*”.

- *clarify which claims are directly derived from the current study, and tone down the ones supported by literature.*

We have taken this into account and made this distinction throughout our revised manuscript.

- *Complement and generalize the bioinformatic analysis of the RNAseq-proteome data (Ref #3, standfirst and pts. 3,4,8; ref#3, pts. 3,4), and integrate the conclusions of the resource part with the in vivo migration assay.*

In the revised manuscript we have better integrated the in vivo migration data with the resource part. Using *in vivo* imaging we found that B16F1 and B16F10 have differential capacity to migrate. In the revised manuscript, we now explain that this is the primary reason why we have analyzed whether the EVs that these cells release have the potential to transfer cargo that can influence migration of the recipient cells.

Moreover, we have compared our EV mass spectrometry data to the data bases Vesiclepedia and Exocarta and have confirmed that over 90% of the most often identified top 100 EV markers are also present in our EV datasets. Additionally, we have compared our EV mass spectrometry data to already published data by other groups (Keerthikumar, Gangoda et al., 2015, Kowal, Arras et al., 2016, Minciacchi et al., 2015b) and identified that that many of their protein makers are indeed enriched in our corresponding EV fractions.

Furthermore, we have generalized the description of our data in the results and discussion section and added: “*Because B16F1 and B16F10 cells have differential migration capacity, we analyzed whether cargo that is specifically loaded in EVs can affect migration. GO term analysis showed that the EV-enriched proteins and RNAs are involved in a variety of biological processes (Figure 4B). The top GO terms are related to processes known to be upregulated during metastases including cell migration, wound healing and morphogenesis (Figure 4B, highlighted in red). The highest functional enrichment is observed for cell surface receptor signaling pathway in the 100K EVs (Figure 4B), Previously, other groups have shown that EV-associated RTKs can be functionally incorporated in the plasma membrane of recipient cells (Al-Nedawi, Meehan et al., 2008, Zhang, Deng et al., 2017). Moreover, >100 proteins that are linked to cell surface receptor signaling pathways were found to be enriched in EVs (Figure 4B, see also Appendix Table S6), implying that EVs could transfer receptor-mediated oncogenic signaling to other cells.*”

In the concluding remarks section, we have added: “*In addition to networks of migration-related RNA and protein, we also found that EVs carry biomolecules with diverse other functions. For instance, EVs of the B16F10 model are enriched for epigenetic and posttranscriptional regulation of gene expression, as well as (m)RNA processing, potentially bringing about diverse effects in cells*

that receive these EVs (Figure EV4B). Moreover, RNA molecules related to glycolysis (Figure 4 and Figure EV5; glycoprotein metabolic process) are enriched in EVs. Interestingly, glycolysis has been shown to be upregulated in migratory cancer cells (Han, Kang et al., 2013, Shiraishi, Verdone et al., 2015). Additionally, enrichment of receptor-mediated signaling in EVs (cell surface signaling pathway, Figure 4B, Figure EV5B and Appendix Table S6) may help to transfer oncogenic properties between cancer cells. Together, The transfer of molecules involved in the above mentioned processes may further amplify the effectiveness of the transferred biomolecules involved in migration.”

- Increase mouse numbers to improve robustness of in vivo metastatic potential data (Figure S6; ref#3, pt.5) or tone down claims on this aspect.

It has not been our intention to overstate our data and we had the impression that we had carefully made our statement by explaining that only one mouse out of 11 mice developed metastatic disease. Although this single mouse pointed to the right direction, in the revised manuscript, we have removed this data.

Comments from the reviewers:

Referee #1:

In this study, Steenbeek et al investigate the reciprocal communication between cells that differ in metastatic potential, but present within the same tumor environment. As a model system, they use the syngenic cell lines B16F1 and B16F10, and determine RNA and protein cargo in extracellular vesicles (EVs) shed by either cell type. The authors find remarkable complexity and differences in EV RNA and protein make-up, dependent on cellular origin and type of EV considered (obtained by differential centrifugation). This is further interpreted modeling these data as functional networks, mainly in categories related to cell motility, resulting in protein and mRNA networks. The authors conclude by showing that EVs originating from B16F10 cells induce enhanced migration by uptake of EVs originating from B16F1 cells.

This study addresses an important question, and technically it is well-performed (in vivo migration assays, proteomics). They identify an amazingly large number of proteins and transcripts in EVs, with an equally large functional diversity. These are important, especially in conjunction with the compositional difference of 16.5k and 100k EVs. However my main concern is a conceptual one, where the authors propose a model for which in the end they do not provide any direct evidence. Specifically, the concept that the authors are aiming to get across (and put in the title) is that the EV-content should be seen as a network, and not merely as a collection of proteins/mRNAs. This may be appealing as a model, however the authors do not really substantiate this beyond drawing connections between proteins/RNAs that co-occur within a certain gene-ontology/functional category. First of all, this is not surprising since this is how GO-terms are defined. Second, a network is defined by interdependencies, meaning that if the network is perturbed it loses functionality. Proving this will involve careful and not necessarily straightforward experimentation, however none of this is attempted and therefore the network-theory remains a theory: it is not conclusive, and there are no data in the manuscript that cannot be explained by a simpler model, namely that the trans-effect is mediated simply by the collection of molecules contained within EVs.

Response:

We would like to thank the reviewer for his/her time and effort to provide us with constructive feedback and helpful suggestions that have improved our manuscript.

We completely agree with the reviewer, and in fact, the points that this reviewer made are exactly what we want to get across to the reader (trans-effect is mediated by the collection of molecules within EVs and not just single or a few of these molecules as currently thought in the field), and we do now realize that our points were not well phrased in the original manuscript. We have tried to get the following points across.

- 1) Several hall-mark papers in the field conclude that single biomolecules (either RNA or proteins) are responsible for the transfer of a phenotype from the donor to the recipient cell (e.g. MET: *Peinado et al Nat med 2012*, miR-105: *Zhou et al Cancer Cell 2014*, PTEN miRNAs: *Zhang et al Nature 2015*, miR-122: *Fong et al NCB 2015*, MIF: *Costa-Silva et al NCB 2015*). These claims are based on the following experiments: knockdown of the biomolecule in the donor cell affects the EV-induced behavioral change in the recipient cells. Although on first sight these experiments sound convincing that a single EV molecule can affect recipient cells, during our project we started to realize that the interpretation of these experiments is more complex. There is overwhelming evidence in the literature that knockdown of key molecules leading to a phenotype in the donor cells (e.g. loss of migration) are associated with global changes in the expression profile of these EV-producing donor cells (just to name a few examples of overwhelming evidence in the literature, see Ding, McConechy et al., 2015, Eraly, 2014, Gerstung, Pellagatti et al., 2015, Verdoni, Aoyama et al., 2008 show how single mutations change the expression profile of 100s-1000s genes). On the contrary, there is to our knowledge no evidence in the literature that alteration of a single gene (leading to a phenotype) does the job all by itself without affecting expression profile of other genes. Taking this knowledge into account, the observed phenotypic change in acceptor cells upon knockdown of single genes in donor cells can well be explained by altered expression profile of the donor cells, and subsequently EV loading (as EVs reflect donor cells) and a completely different EV-induced phenotype in the recipient cell.

However, based on these hallmark papers (which we think are still very important and insightful because they demonstrate the extraordinary role of EVs), the dogma in the field is that EV-induced effects are mediated by single (or a few) molecules loaded in EVs. With our manuscript, we try to change this dogma, and let the field realize that EVs contain many different biomolecules. Thereby, it is far more likely that the trans-effect is mediated by the collection of all biomolecules with a similar biological function (as deduced from network-based analyses and GO mining, see our response to major point 4) than just a few specific ones. Although we do realize that it may be impossible to provide evidence for our model (as this reviewer cleverly points out), we do think that the field should realize that the current dogma may be far more unlikely than the model we propose in our manuscript. Below, we further explain how we address this important point in the revised manuscript.

- 2) We agree that signaling networks require the interdependencies that this reviewer describes and do agree that we have miss-used this term (which we have corrected in the revised manuscript (see below)).

Before we go into the networks, it is important to realize that, similar to the physiological situation, we investigate EV exchange between tumor cells that have a common origin and therefore, molecular differences are small but instrumental. From our comparison of the tumor cells (Fig EV4), we found that all the signaling “networks” that are loaded in the B16F10 EVs are already present in the recipient B16F1 cells, but the relative abundance of each node (protein in the network) differs (Fig 3). Therefore, the cargo of EVs produced by B16F10 does not “transplant” completely new “networks” in the recipient B16F1 cells (which indeed would make each node interdependent), but instead amplifies each existing node in the network already present in the cell. As shown by others (e.g. (Condeelis et al, 2005; Wyckoff et al, 2007), amplifying multiple nodes in the same pathway will have two effects: first, the output of the pathway can be much stronger when two nodes are hit simultaneously, and second, not every recipient cell needs to receive the same cargo (which may vary between single vesicles) to have the same signaling pathway amplification. However, we agree with this reviewer that complete signaling networks may not be the correct term, since our “networks” do not represent full pathways that are reliant on interdependencies. Therefore, in the revised manuscript, we have rephrased the title, abstract, conclusions and discussion to explain that the EV molecules that amplify numerous nodes in the signaling network will have a concerted effect. Moreover, we referred to papers that explain that amplification of network nodes amplifies the output of the signaling networks.

Taking the reviewers suggestion into account, we have changed the title of our manuscript into: “*In vivo tumor cells phenocopying migratory behavior exchange numerous nodes of signaling networks via EVs*”. We have updated the abstract by adding: “*we show that EVs shed by these clones into the tumor microenvironment contain biomolecules of numerous nodes of signaling networks involved in cellular processes including migration*” and

“...suggest this is mediated by a diverse range of EV biomolecules that concertedly amplify numerous nodes within signaling networks.” In the concluding remarks on page 17 we have added: “EV cargo does not “transplant” completely new networks into the recipient cells, but instead amplifies existing nodes in the network already present in the cell.”

Other remarks:

1. The authors do not discuss how they envision mRNA to transmit function (if any) in recipient cells. Are RNAs stable throughout the process of EV transfer, including internalization? Are they translated?

Response:

We apologize for not describing in the introduction the current status of the literature sufficiently enough, since the functional transfer of EV-RNAs is extensively studied by other groups. In an early landmark study by the Lotvall lab, it was shown that functional mRNA loaded in EVs released from mouse cells can be taken up by human mast cells and importantly, that the uptake of this mRNA leads to translation into functional proteins (Valadi et al. 2007, Nature Cell Biology). Furthermore, multiple studies have shown that EV-loaded miRNAs can suppress their target genes in recipient cells (for example Hergenreider, 2012, Nature Cell Biology, Fong 2015, Nature Cell Biology, Zhang et al. 2015, Nature). Additionally, another reporter system that relies on RNA transfer by EVs shows functional transfer of EV associated GlucB mRNA (Lai, et al., 2015, Nature Communications). We showed in human xenografts (Zomer et al., Cell 2015; Zomer et al., Nat Prot, 2016) and now in this manuscript in a mouse allograft system that Cre activity (mostly due to mRNA, see Zomer et al, Cell 2015) is functionally transferred. In the revised manuscript we wanted to make the reader aware of this important literature have adapted the introduction to better describe these important hallmark findings that illustrate the stability and functionality of EV-associated (m)RNA. At page 3 and 4 of the introduction, we have added: “EV-associated biomolecules such as EV-RNA are stable in EVs and functional upon delivery into recipient cells. For example upon EV uptake, vesicular mRNA is translated into functional proteins (Valadi, Ekstrom et al., 2007), and vesicular miRNAs suppresses target genes in recipient cells (Fong, Zhou et al., 2015, Hergenreider, Heydt et al., 2012, Zhang, Zhang et al., 2015). Moreover, EV RNA-based reporter systems have confirmed the transport of functional mRNA for Cre (Ridder, Sevko et al., 2015, Zomer, Maynard et al., 2015, Zomer, Steenbeek et al., 2016) or GlucB (Lai, Kim et al., 2015) into recipient cells.”

2. Figure 1E: it is surprising that EVs remain intact after uptake. Have the authors investigated their ultimate destination, and where/when they dissipate (if at all)?

Response:

The ultimate destination and the uptake mechanisms of EVs are heavily investigated by various groups in the field. The (temporary) persistence of fluorescently-labelled EVs in a recipient cell is in line with other studies that show that fluorescently-labeled EVs stay intact after uptake through endocytosis or micropinocytosis (Costa-Silva et al., Nature Cell Biology, 2015, Fong et al., Nature Cell Biology, 2015. Christianson et al., PNAS 2013). Though exact mechanisms for EV uptake are not clear to date, it is generally perceived that intracellular membrane fusion mediates endosomal/lysosomal escape of EVs, releasing EV content into the cytosol (as reviewed by Ruivo et al., Cancer Res. 2017, Mulcahy et al., 2014 J Extracell Vesicles, Raposo et al., J Cell Biol 2013). Interestingly, as we observe a low number of eGFP⁺ reporter⁺ cells *in vitro* as shown in Figure 1 G-I of our manuscript, the uptake and particularly subsequent functional release of EV content is likely to be a more efficient process *in vivo* than *in vitro*.

To clarify this, in the revised manuscript (page 3 and 4) we have expanded in the introduction on the literature that describes EV uptake and the release of the EV-content into recipient cells such as EV-RNA. We have added: “For example upon EV uptake, vesicular mRNA is translated into functional proteins (Valadi et al. 2007, Nature Cell Biology), and vesicular miRNAs suppresses target genes in recipient cells (Hergenreider, 2012, Nature Cell Biology, Fong 2015, Nature Cell Biology, Zhang et al. 2015, Nature). Moreover, EV RNA-based reporter systems have confirmed the transport of functional mRNA for Cre (Ridder et al, 2015, Zomer et al., Cell 2015; Zomer et al., Nat Prot, 2016) or GlucB (Lai, et al., 2015, Nature Communications) into recipient cells”.

3. Figure 3: the authors restrict themselves by considering proteins/transcripts involved in cell migration. However, the number of proteins in EVs is so high, and their functionality so diverse, that multiple other cellular processes are ignored that could be equally important to effectuate a trans-effect to recipient cells. So if the authors want to propose EVs to represent networks, Fig 3 (and 5) represent only a small portion of such a network.

Response:

We agree with the reviewer that we have mainly focused on the differences in cell migration. As the reviewer correctly noticed, we also observed vesicular RNAs and proteins related to other processes. For example, differential expression analysis between the B16F1 and B16F10 model shows that RNAs related to “epigenetic regulators of gene expression”, posttranscriptional regulators of gene expression and “RNA processing” are highly enriched in EVs of the B16F10 model (see also Figure S4B in the original manuscript). In the original manuscript, we have tried to explain this to the reader by showing the GO-term analysis in Figure 4 and Figure S4. Moreover, in the concluding remarks of our original manuscript we have tried to emphasize this point by stating that “*In addition to networks of migration-related RNA and protein, we also found that EVs carry biomolecules with diverse other functions. For instance, EVs of the B16F10 model are enriched for epigenetic and posttranscriptional regulation of gene expression, as well as (m)RNA processing, potentially bringing about diverse effects in cells that receive these EVs (Figure S4B). This may further amplify the effectiveness of the transferred biomolecules involved in migration.*”

We now realize that it may help to get these points better across to the reader by discussing this data more prominently in the manuscript, and in the revised manuscript discuss in the results section that EVs are specifically enriched for a variety of processes, also including cell migration. Moreover, in the closing remarks we now also discuss the potential role of other EV cargo related to glycolytic processes and receptor-mediated signaling.

In the results section on page 13, we have rewritten the generalized description of our findings and added: “*GO term analysis showed that the EV-enriched proteins and RNAs are involved in a variety of biological processes (Figure 4B). The top GO terms are related to processes known to be upregulated during metastases including cell migration, wound healing and morphogenesis (Figure 4B, highlighted in red). The highest functional enrichment is observed for cell surface receptor signaling pathway in the 100K EVs (Figure 4B). Previously, other groups have shown that EV-associated RTKs can be functionally incorporated in the plasma membrane of recipient cells (Al-Nedawi, Meehan et al., 2008, Zhang, Deng et al., 2017). Moreover, >100 proteins that are linked to cell surface receptor signaling pathways were found to be enriched in EVs (Figure 4B, see also Appendix Table S6), implying that EVs could transfer receptor-mediated oncogenic signaling to other cells.*”

In the concluding remarks section on page 17, we have added: “*In addition to networks of migration-related RNA and protein, we also found that EVs carry biomolecules with diverse other functions. For instance, EVs of the B16F10 model are enriched for epigenetic and posttranscriptional regulation of gene expression, as well as (m)RNA processing, potentially bringing about diverse effects in cells that receive these EVs (Figure EV4B). Moreover, RNA molecules related to glycolysis (Figure 4 and Figure EV5; glycoprotein metabolic process) are enriched in EVs. Interestingly, glycolysis has been shown to be upregulated in migratory cancer cells (Han, Kang et al., 2013, Shiraishi, Verdone et al., 2015). Additionally, enrichment of receptor-mediated signaling in EVs (cell surface signaling pathway, Figure 4B, Figure EV5B and Appendix Table S6) may help to transfer oncogenic properties between cancer cells. Together, The transfer of molecules involved in the above mentioned processes may further amplify the effectiveness of the transferred biomolecules involved in migration.*”

4. Figure 3B: what do these 'associations' mean, how are they defined? Direct interactions, enzyme/substrate, (transcriptional) regulation, co-occurring in the literature? Authors need to clarify this - for instance bottom of page 9: 'the highest enrichment of migration-related RNA and protein is generally observed in EVs, suggesting that EV-mediated transfer of these molecules could result in a concerted action in recipient cells'. This really depends what the 'association' between these genes/proteins indicate. If it is transcriptional regulation of one by the other, transfer of both proteins does not have a meaning in the sense of a concerted action.

Response:

We completely agree with the reviewer that the associations we used in our original manuscript should be clearly defined and apologize for being insufficiently clear about this. First, we have better clarified what interaction means (see below), and second, what the evidence is for these interactions (see below).

First the definition of interactions. The interactions used in the original manuscript represent protein-protein associations exported from the quality-controlled STRING database (Szklarczyk et al., Nucleic Acids Res., 2017). This analysis was performed online at the following website: string-db.org. In this database (STRING), interactions are defined as: “associations (that) are meant to be specific and meaningful, i.e. proteins jointly contribute to a shared function; this does not necessarily mean they are physically binding each other”. The reviewer is correct that this also includes enzyme/substrate, (transcriptional) regulation, and other indirect interactions between proteins in the same network that influence each other. In the revised manuscript, we now define the interaction at page 12: *“These interactions are defined as “associations (that) are meant to be specific and meaningful, i.e. proteins jointly contribute to a shared function; this does not necessarily mean they are physically binding each other”. These interactions includes enzyme/substrate, (transcriptional) regulation, and other indirect interactions between proteins in the same network that influence each other.”*

Second, the evidence for the interactions: in the original manuscript, these interactions from STRING also included predicted interactions such as gene neighborhood, gene fusions, gene co-occurrence, and other interactions based on text-mining, co-expression and protein homology. We now realized, and agree with the reviewer, that not all interactions have the same importance (for example, importance of experimentally determined is higher than importance of text-mined interactions). Therefore, in the revised manuscript we used the STRING database to only select for interactions that are based on experimental evidence. To this end, we have re-analyzed the evidence for the interactions and have updated Figure 3B, C and Figure 5B to only represent edges supported by evidence from experimentally curated databases (Biocarta, BioCyc, Gene Ontology, KEGG and Reactome) and experimentally determined interaction databases (DIP, BioGRID, HPRD, IntAct, MINT, and PDB).

This new analysis confirms that most interactions originally identified are experimental and/or known interactions, confirming that the identified biomolecules in Figure 3 and Figure 5 can act as a molecular machinery. To further support our conclusion that we have indeed identified a migration signaling network, we have also included the raw edge evidence in the supplemental “SourceDataForFigure3” and “SourceDataForFigure5”.

To make the network evidence of the revised figures clear to the reader, in the results and conclusion section on page 12 we have added: *“We only included protein-protein interactions that are experimentally validated and/or annotated in curated databases (Figure 3A and Materials and Methods)”*

Additionally, we mention the EDGE evidence in the figure legends of Figure 3 and 5: *“differential expressed genes or proteins plotted on the interaction network of cell migration, only representing experimentally determined interactions and interactions from curated databases”*. Moreover, we have updated the material and methods section on page 29 with: *“Only edges representing evidence of known interactions from curated databases (Biocarta, BioCyc, Gene Ontology, KEGG and Reactome) and experimentally determined interactions (DIP, BioGRID, HPRD, IntAct, MINT, and PDB) were preserved with protein homology to other species”*

5. Related to the previous point, authors state (page 9-10): *‘The highest functional enrichment is observed for cell surface receptor signaling pathway in the 100K EVs, implying that EVs could transfer receptor-mediated oncogenic signaling to other cells’. This is somewhat of a simplistic view, since RTKs only function if they are properly positioned in the membrane, to sense growth factors and induce a response via second messengers etc. I.e. they only function when the whole machinery is in place, for which no evidence is provided.*

Response:

As we explained above, similar to the physiological situation, we investigate EV exchange between tumor cells that have a common origin and therefore, molecular differences are small and all the machinery nodes are loaded in the B16F10 EVs are already present in the recipient B16F1 cells, but the relative abundance of each node differs (Fig 3). With other words, the whole machinery is already in place. Moreover, it is important to note that many labs have studied the transfer of RTKs and have shown that EV-associated RTKs are functionally incorporated in recipient cells. For instance, EV-derived EGFR or EGFRvIII are functionally incorporated in the plasma membrane of recipient cells (H. Zhang et al, 2017, Nature communications, Al-Nedawi K, 2008, Nature Cell Biology). Nevertheless, we also initially thought that RTKs only get activated if they sense growth factors, until one of our nearby colleagues told us that up regulation of the receptor can already lead to auto activation without growth factors (probably due to stochastic auto-cross phosphorylation). He explained that in cancer, cell surface receptors like EGFR do not need to be mutated to influence tumor growth, but amplification of the wild-type receptor also promotes tumor growth (e.g. Talasila K. et al., Acta Neuropathol. 2013, William D. et al., PLoS One, 2017, Ciardiello F., and Tortora G, NEJM, 2008). Additionally, we have further investigated the identified proteins linked to “cell surface receptor signaling pathway” and found that there are over 100 proteins enriched linked to cell surface receptor signaling pathway and that these also include non-RTK proteins. To better explain this to the reader in the revised manuscript, we have included these references (page 13) and included the data in Appendix Table S6: *“The highest functional enrichment is observed for cell surface receptor signaling pathway in the 100K EVs (Figure 4B), Previously, other groups have shown that EV-associated RTKs can be functionally incorporated in the plasma membrane of recipient cells (Al-Nedawi, Meehan et al., 2008, Zhang, Deng et al., 2017). Moreover, >100 proteins linked to cell surface receptor signaling pathways were found to be enriched in EVs and as cell surface receptors and also downstream effectors are enriched in EVs (Figure 4B, see also Appendix Table S6), implying that EVs could transfer receptor-mediated oncogenic signaling to other cells.”*

6. Page 11-12: Here the authors seem to perform an in-silico analysis of what proteins/RNAs are transferred between B16F1 and B16F10. It is unclear what can be concluded without experimentally demonstrated that the mentioned molecules are indeed transferred, and have a function.

Response:

We apologize for not being clear about this analysis. In the EV field, a biological molecule loaded in EVs is considered important if it is relatively abundant in EVs compared to the donor cell. However, a biological molecule that is specifically sorted in EVs but also very abundant in the recipient cell is not likely to add much to the recipient cell (as an analogy: an extra glass of water in the sea would not have an effect). By contrast, other vesicular biomolecules that may not be specifically sorted into EVs can likely have a large effect if this molecule is at low abundance in the recipient cell (as an analogy: a glass of water in the desert will have a large effect). Therefore, solely studying the content of EVs does not show which biomolecules will have a strong effect in the recipient cells. Accordingly, it is required to take into account the abundance of each biomolecule in the recipient cell. Since we are strongly convinced that this analysis helps the reader to understand how to interpret the importance of each vesicular biomolecule for inducing a phenotypic change in the recipient cell, we prefer to keep this analysis in the revised manuscript.

We do now realize that we did not get our point well across to readers. Therefore, we have rephrased this part of the revised manuscript and explain better why we perform this analysis. In the revised manuscript we have added to page 14: *“Despite specific loading of some of the vesicular RNA and proteins, they may not all have the same effect in the recipient cells. For example, a biological molecule that is specifically sorted in EVs but also very abundant in the recipient cell is not likely to add much to the recipient cell. By contrast, other vesicular biomolecules that may not be specifically sorted into EVs can likely have a large effect if this molecule is at low abundance in the recipient cell. Therefore, solely studying the content of EVs does not show which biomolecules will have a strong effect in the recipient cells”*.

We appreciate the comment of this reviewer that this analysis only provides a theoretical weight to each biomolecule that can never be experimentally proven. To clarify this to the reader, in the revised manuscript we emphasize this point by stating that this is a theoretical analysis 14-15: *“In an attempt to provide theoretical weight for the most effective vesicular RNA and proteins, we*

identified...”, and “This shows that theoretically, both B16F1 and B16F10 EVs”, and “B16F10 EVs theoretically transfer more positive regulators of cell migration”.

7. Fig 6CD: have the observed effects been corrected for the number of EVs that have been taken up per cell?

Response:

The number of EVs that is taken up by recipient cells is an intriguing question and still heavily debated in the field. Unfortunately, our Cre-LoxP system, as any other currently available technique based on the transfer of reporter molecules by EVs, cannot answer this question. Although we think that this is an important question to address, in this manuscript we feel it is beyond the scope where our main aim is to show that EVs are not just transferring single (or a few) different molecules, but instead molecules will combined have an action.

8. Page 14: 'tumor cells transfer networks of interconnected RNAs and proteins'. What does 'interconnected' mean? This relates to my main concern: is it proteins in the same pathway, or the RNAs that encode them? This only makes sense when authors assume that an entire pathway or other functional module is 'transplanted' into a recipient cell. This really cannot be concluded from the motility experiments (Fig 6).

Response:

As explained above, it is important to realize that the signaling “networks” that are loaded in the B16F10 EVs are already present in the recipient B16F1 cells, but the output (migration) of these “networks” is in the B16F1 cells not as strong as in the B16F10 cells. Therefore, the cargo of EVs produced by B16F10 does not “transplant” new “networks” in the recipient B16F1 cells (which would make each node interdependent), but instead amplifies each existing node in the network already present in the cell. As shown by others (e.g. Condeelis et al, 2005; Wyckoff et al, 2007), amplifying multiple nodes in the same pathway will have two effects: first, the output of the pathway can be much stronger when two nodes are hit simultaneously, and second, not every recipient cell needs to receive the same cargo (which may vary between single vesicles) to have the same signaling pathway amplification. However, we agree with this reviewer that signaling networks may not be the correct term, since our “networks” do not have interdependencies as some linear signaling pathways. Therefore, in the revised manuscript, we rephrased the title, abstract and conclusions that the concerted action of all EV molecules and amplification of numerous network nodes will most likely mediate the effect, as we have also noted at the major comment of this reviewer. Moreover, we speculated in the discussion on page 18 on how multiple molecules in the same pathways can help to robustly transfer a phenotype: “*In order to become motile in the primary tumor, cancer cells upregulate multiple parallel networks of signaling pathways involved in migration (Figure 3B, C) (Wang, Goswami et al., 2004, Wang, Goswami et al., 2005, Wyckoff, Wang et al., 2007). Importantly, analysis of migratory cells isolated from tumors with different genetic origins showed that upregulation of different genes in the same pathway can be altered in different cells to achieve the same migratory behavior (Condeelis, Singer et al., 2005, Wyckoff et al., 2007). Our data shows that the pool of EVs released by tumor cells transfer networks of interconnected RNAs and proteins. This data illustrates that it may not be the upregulation of the level of one particular RNA or protein that mediates the phenotypic change of recipient cells, but it could be the upregulation of the activity of migratory pathways as a whole.*”

9. More generally speaking, there is a disconnect between proteome/sequencing data and functional analysis. Strictly speaking, Fig 6 could also be figure 1 since none of the acquired knowledge was necessary or helpful to do the cell migration experiment. Even more, it remains unclear to which EV-type the observed effect can be ascribed (16.5k, 100k or something else), leave alone to which protein/RNA they contain.

Response:

We argue that it is the presumable concerted action of vesicular biomolecules (as deduced from network-based analyses and GO mining, regardless of the EV-type) that mediates the phenotypic switch in the recipient cells. Therefore, the question whether different types of EVs can potentially have different effects is beyond the scope of this manuscript (though we agree it is an interesting question that is currently addressed by big studies in other labs in the field).

10. The final conclusion is very sketchy where authors talk about possible consequences of changes in EV-cargo due to mutations or otherwise, however this remains very speculative without having data or tools to observe and quantify these effects.

Response:

As explained above in the response the general points, these final conclusions are based on an overwhelming evidence in the literature that mutations and knockouts of a single gene (especially when inducing a phenotype) can lead to broad differences in gene expression, with a described change of 100s-1000s of differentially transcribed genes (for example: Ding, McConechy et al., 2015, Eraly, 2014, Gerstung, Pellagatti et al., 2015, Verdoni, Aoyama et al., 2008). To explain this to the reader, in the revised manuscript, we have added references to statements that are based on data from literature at page 18: “Only if the whole loading of EVs is changed, for example due to a changing expression of hundreds to thousands of genes in the donor cell upon a gene alteration (Ding, McConechy et al., 2015, Eraly, 2014, Gerstung, Pellagatti et al., 2015, Verdoni, Aoyama et al., 2008), the response of the recipient cell will change.”

Referee #2:

The article by Steenbek et al provides a very thorough comparative study of the protein and mRNA composition of EVs of different biophysical characteristics, isolated in vivo from a more and a less metastatic subclone of a given tumor cell line. Refined bioinformatics analyses of these extensive array of data leads to the hypothesis that EVs can transfer clusters of RNA and proteins acting in the same physiological pathways, to promote e.g. migration of less migratory cells. The authors validate this hypothesis by showing enhanced in vivo migration of cells that have captured pro-migratory EVs.

This article follows on a very elegant previous study of this group (Zomer, Cell 2015), where the authors had implemented use of the Cre/LoxP system to evidence functional transfer of Cre mRNA between two human tumor cell lines. The experimental system used here is more relevant to the clinical situation, as it uses two subclones of a tumor cell line, rather than two completely different tumor cell lines, thus mimicking the intrinsic tumor heterogeneity of a tumor. It is also implemented in fully immunocompetent mice, instead of relying of injection of human tumor cells in nude mice.

The article provides a large range of very valuable information for the EV and cancer fields, especially the extensive proteomic and transcriptomic analyses of EV subtypes from different sources. It confirms and extends recent studies from other groups highlighting the functional relevance of different subtypes of EVs, not only "exosomes"! It also provides an important observation that Cre transfer is not efficient in vitro, and only observed in vivo (figure 1). It is certainly of high value as a resource article.

Response:

We would like to thank the reviewer for his/her time and effort to provide us with constructive feedback and helpful suggestions for additional experiments to improve our manuscript.

I would suggest an additional experiment, that I think is missing in the current manuscript to properly interpret the migration data upon EV transfer: quantification of Cre-mediated recombination (as in figure 1H) and resulting migration (as in Fig 6) in reporter cells co-injected with Cre-donor cells of the same subclone, ie B16F1-Cr⁺ B16F1-reporter⁺. Indeed, the observed increased migration of B16F1-reporter upon capture of EVs from B16F10-donors is significant but not major, and B16F1-derived EVs also contain a majority of pro-migratory networks (Fig5B), even if slightly more anti-migratory networks than B16F10-derived EVs. Thus, can the authors demonstrate that capture of these F1 EVs does not induce as much or as statistically significant migration in F1 cells as transfer of F10-EVs, which would support their model of EV-mediated phenocopy, or not (as much migration in both cases), in which case the phenocopy model does not hold. However, even if the latter result is observed, the major importance and value of the article as resource remains.

Response:

We agree with the reviewer that investigating the role of EV transfer within the same model will help us gain more insight into the role of pro-migratory networks in EVs. During the revision time of the current manuscript, we have performed this experiment and studied the requested effect within one tumor type, i.e. from B16F1 to B16F1. We mixed B16F1 Cre⁺ CFP cells with B16F1 reporter⁺ cells, and studied the differential behavior of DsRed⁺ and eGFP⁺ reporter⁺ cells. This shows that the B16F1 eGFP⁺ cells do not have a higher migration speed (fold change 1.03) as was predicted by this reviewer. We have included these important findings in updated Figure 6 and describe this data at page 16 of the revised manuscript: “By contrast, when the B16F1 cells take up EVs produced by B16F1 cells, the migration speed is not enhanced (Figure 6E).”

I also have minor comments, that could be taken into account to improve the manuscript.

1) Characterisation of the B16F10 and F1 models and their growth in vivo, mentioned in the beginning of results, should be shown as supplementary data.

Response:

We have observed that the growth rate of these tumor models *in vivo* is comparable for the B16F1 and B16F10 models. However, since our manuscript is already data dense, we prefer to only add data that directly support the main message of the paper. If this reviewer feels that it will add to our story, we are willing to add this data to the revised manuscript.

2) Figure 1i should be displayed as dot plots and not histograms, to show distribution of the eGFP⁺ cell number in individually analyzed tumors/mice.

Response:

In the revised manuscript, we included the dot plots instead of representing the data as histograms in Fig 1i.

3) I did not understand exactly the analysis displayed in figure 5: I suppose that the enrichment on EVs over "putative recipient cells" (figure legend) was searched for in the proteomic and transcriptomic data generated for the rest of the article, and not on novel experiments where, for instance, proteomic/transcriptomic analysis would have been done on cells that had captured or not EVs and thus recombined or not, but I am not entirely sure. Can the authors try to explain better in the text their reasoning in this part of the analysis? For instance, in p11, the sentence "we selected for RNA and proteins enriched in EVs to recipient cells" is not very meaningful.

Response:

We apologize to the reviewer for being unclear about this experiment. Let us first explain the reason for this analysis. A biological molecule loaded in EVs is not per definition important if it is specifically sorted into EVs. For example, a biological molecule that is specifically sorted in EVs but also very abundant in the recipient cell is not likely to add much to the recipient cell. By contrast, other vesicular biomolecules that may not be specifically sorted into EVs can likely have a large effect if this molecule is at low abundance in the recipient cell. Therefore, solely studying the content of EVs does not show which biomolecules will have a strong effect in the recipient cells. Accordingly, it is required to take into account the abundance of each biomolecule in the recipient cell, and that is what we did in Figure 5. Since we are strongly convinced that this analysis helps the reader to understand how to interpret the importance of each vesicular biomolecule for inducing a phenotypic change in the recipient cell, we prefer to keep this analysis in the revised manuscript. We agree with the reviewer that we could have explained our reasoning for this experiment better, and we have added to page 14 of the revised manuscript: “Despite specific loading of some of the vesicular RNA and proteins, they may not all have the same effect in the recipient cells. For example, a biological molecule that is specifically sorted in EVs but also very abundant in the recipient cell is not likely to add much to the recipient cell. By contrast, other vesicular biomolecules that may not be specifically sorted into EVs can likely have a large effect if this molecule is at low abundance in the recipient cell. Therefore, solely studying the content of EVs does not show which biomolecules will have a strong effect in the recipient cells”. We do agree with the reviewer that this analysis only provides a theoretical weight to each biomolecule that can never

be experimentally proven, and in the revised manuscript we have emphasized this better to the reader at page 14-15: “*In an attempt to provide theoretical weight for the most effective vesicular RNA and proteins, we identified...*”, and “*This shows that theoretically, both B16F1 and B16F10 EVs*”, and “*B16F10 EVs theoretically transfer more positive regulators of cell migration*”.

4) *the proteomic data comparing EVs of different natures (16,5g vs 100,000 g) could have been valuably discussed in view of other recent studies comparing also different EV subtypes: what parts of the authors' results are consistent with these previous studies (eg Keerthikumar, Oncotarget 2015; Minciacchi Oncotarget 2015; Willms Sci Rep 2016; Kowal PNAS 2016; Haraszti JEV 2016)?*

Response:

We agree with the reviewer that it is interesting to compare our data to other recent studies. For the revised manuscript, we have compared our data to the Exocarta and Vesiclepedia databases and we compared our data to Keerthikumar, oncotarget, 2015, Minciacchi, oncotarget, 2015 and Kowal, PNAS, 2016 as suggested by the reviewer. We have included these analysis on page 9-10 of the revised version of the manuscript: “(..) *from the top 100 most often published EV makers described in ExoCarta (Haraszti, Didiot et al., 2016, Keerthikumar, Chisanga et al., 2016) and Vesiclepedia (Kalra, Simpson et al., 2012), more than 90% of these EV markers were also present in all our replicates of the 16.5K and 100K EV fractions (Appendix Table S2 and S3). Interestingly, we observe that the EVs in different fractions have differential cargo. Proteins shown to have high abundance in DU145 cell-derived 10K EVs (Minciacchi et al., 2015b) show strong enrichment in our 16.5K EVs but not in our 100K EVs, which may suggest the presence of specific markers per type of EV (e.g. HSPD1, HSPA9, MDH2, see also Appendix Table S4). A recent study showed a lack of exclusive marker genes for exosomes and microvesicles when EVs from different cell lines were compared (Haraszti et al., 2016) but in line with our findings, this study found that 100K exosomes were enriched for proteins of receptors and cell adhesion, whereas 10K MVs were enriched in endoplasmic reticulum and mitochondrial proteins (see below). A few other studies have claimed the existence of markers that exclusively differentiate between exosomes or microvesicles (Keerthikumar, Gangoda et al., 2015, Kowal, Arras et al., 2016, Minciacchi et al., 2015b) (See also Appendix Table S5). Our data shows that many of these makers are indeed enriched in our 100K EV (e.g. VPS36, ITGA5, TSG101) or 16.5K EV (e.g. GLS, TUFM, ETFA) fractions (see also Appendix Table S4). However, these markers are not exclusively present in either one of the fractions, which may be explained by the fact that differential centrifugation does not yield pure fractions.*”

Of note, Lamp2 is presented as a "typical exosome marker" "specifically enriched in the 100k fraction" (text p8), which does not match figure 2D (Lamp2 is more abundant in 16;5 than 100K) and the literature (Lamp2 is not consistently found associated to EVs, it is even described as excluded from exosomes in some studies).

Response:

We agree with the reviewer that whilst other markers have the highest enrichment in 100K EVs, this is not the case for LAMP2, which has similar expression in the 16.5K and 100K Fraction. We have reviewed the presence of LAMP2 in EV preparations in other studies, and agree with the reviewer that Lamp2 is not consistently found associated to EVs. Therefore in the revised manuscript, we have removed LAMP proteins as a classical EV marker from Figure 2D.

5) *the authors should check carefully all figures, for possible mistakes in symbols or presentation: example in fig3c, where panel "positive regulator/B16F1 specific" shows pie-charts of Adam10 and Itgb3 displayed upside-down (Cell at bottom), and/or maybe left-right inversion?*

Response:

Thanks for pointing out these typos. For the revised manuscript, we have thoroughly checked all figures for other typos and correct where required.

In Figure 4B, panel "Cells vs 100K, B16F1", cells are on the right and 100K on the left, whereas in all other panels of this comparison "EVs vs donor cell, .. gene..." cells are positioned on the left.

Response:

We apologize for the inconsistency and will correct this in the revised manuscript.

Also scales of "-log10 corrected p-values" are -2... -8 or -5... -20 in some graphs, and +5... +20 or +2... +10 in others: is it normal?

Response:

Thanks again for pointing this out. We have checked the p-values and we agree that there is an inconsistency between the different figures. In the revised manuscript, we have corrected the axis of the GO-term enrichment scores to represent $-\log_{10}$ p-values of positive numbers on the x-axis.

6) although nice and providing interesting resource data, the protocol used to isolate EVs from in vivo grown tumors is not as refined as that used by Vella, JEV 2017, as it does not involve any further separation of EV subtypes in a density gradient. Hence, the EVs recovered are mixed and could also include some cell fragments, the authors should acknowledge that.

Response:

Yes, we agree with the reviewer, and in the revised manuscript we have acknowledged these facts. In the concluding remarks on page 16, we have added: "(...) it should be realized that these two EV populations may still contain multiple EV types or non-detectable cell fragments that are released upon isolation procedure (Figure 2F, G). EV subtypes could be potentially further fractionated using density gradient centrifugation (Vella et al., 2017) or antibody capture (Kowal et al., 2016)"

7) It would be really interesting to determine (in a future work) if the negative and positive networks of cell migration found in the B16F1 EVs are in fact present in different EV subtypes, which, if differentially captured by target cells, would then efficiently induce either a pro-migratory, or an anti-migratory effect, rather than the suggested mixed induction of contradictory functions.

Response:

We fully agree with the reviewer that this would be an outstanding question for future work. We have mentioned this in the discussion of the revised manuscript. On page 18 of the revised manuscript, we have added: "future studies will have to identify if and how the total EV cargo is divided over individual EVs of the same and different subtypes."

Referee #3:

Steenbeek et al. have studied extracellular vesicles (EVs) as regulators of cancer cell migratory behavior. Their manuscript contains a very detailed characterization of the contents of the EVs. In particular, these EVs are isolated from tumors derived from two sister cell lines - a metastatic and a non-metastatic cell line - and not, as is otherwise often the case, from cells in culture. Thus, this characterization has merit. However, the information gained from the analysis of the EV contents is entirely descriptive. The functional data that is included in the paper is not connected to the characterization of the EVs. Furthermore, the functional data don't add anything substantially new to the functional data previously published by the same group (Zomer et al., 2015). There is no validation convincingly showing that any of reported "omics-based" findings has functional consequence. Regrettably, the EV field doesn't really have the tools to effectively test the premise of the paper, namely that transfer of a network of molecules in EVs - from one cancer cell population to another - is a means for transfer of the ability to metastasis between cancer cell populations.

Response:

We would like to thank the reviewer for his/her time and effort to provide us with feedback and suggestions that helped to significantly improve our manuscript.

We would like to explain in more detail that this manuscript does add considerable novelty to our previous Cell paper from 2015 by Zomer et al.

- 1) The Cell paper was a proof of concept study, where we show that the highly aggressive basal breast cancer cell line MDA-MB-231 can phenocopy its metastatic behavior to the more benign luminal A breast cancer line T47D when co-transplanted into immune-

deficient mice (Zomer et al, 2015). Although these proof-of-principle experiments illustrate the potential of EV exchange to phenocopy differential behavior, they do not model EV exchange in tumors with cells from the same sub-type (and therefore more subtle differences. In the current manuscript (and in contrast to the Cell paper), we now also show that the results hold true for a more physiological (two sister lines) and an immune-competent setting. This is a substantial and critical addition to the functional data that we have previously published in Cell.

- 2) As this reviewer correctly points out, it would be important to interfere with the EV transfer to illustrate that the “omic-based” findings have a functional consequence. For example, it would be effective to inhibit the release of EVs in the donor cells, and test how the lack of transfer of a network of molecules affects the behavior of the recipient cell. As the reviewer also correctly points out, the EV field doesn’t really have the tools, and moreover, if they were available, it would be difficult to interpret the data (it would lead to many non-EV mediated changes). However, our Cre-LoxP system is an alternative approach which provides exactly the same information: studying recipient cells that do not take up EVs (which would be the same as recipient cells that do not take up EVs upon interference with the release of EVs of donor cells). In our experiments, we ‘mark’ this population of cells with DsRed, serving as an internal control that is requested by this reviewer. Importantly, our approach has the advantage that it does not necessarily require manipulation of the donor cells to distinguish between recipient cells that have, or have not, taken up EVs, and therefore we can exclude potential artifacts induced by donor cell manipulation. Since we compare the behavior between cells that have and have not taken up EVs, our data provides exactly the information and validation that this reviewer suggests.
- 3) A major point we want to get across in our manuscript is that EV-induced changes in cell behavior are mediated by transfer of a collection of molecules with presumable concerted action as deduced from network-based analyses and GO mining, and not just one single or a few of these molecules. This is an important message since several hall-mark papers in the field conclude that single biomolecules (either RNA or proteins) are responsible for the transfer of a phenotype from the donor to the recipient cell (e.g. MET: Peinado et al Nat med 2012, miR-105: Zhou et al Cancer Cell 2014, PTEN miRNAs: Zhang et al Nature 2015, miR-122: Fong et al NCB 2015, MIF: Costa-Silva et al NCB 2015). These claims are based on the following experiments: knockdown of the biomolecule in the donor cell affects the EV-induced behavioral change in the recipient cells. Although on first sight these experiments sound convincing that a single EV molecule can affect recipient cells, during our project we started to realize that the interpretation of these experiments is more complex. There is overwhelming evidence in the literature that knockdown of key molecules leading to a phenotype in the donor cells (e.g. loss of migration) are associated with global changes in the expression profile of these EV-producing donor cells (just to name a few examples of overwhelming evidence in the literature, see Ding, McConechy et al., 2015, Eraly, 2014, Gerstung, Pellagatti et al., 2015, Verdoni, Aoyama et al., 2008 show how single mutations change the expression profile of 100s-1000s genes). On the contrary, there is to our knowledge no evidence in the literature that alteration of a single gene (leading to a phenotype) does the job all by itself without affecting expression profile of other genes. Taking this knowledge in account, the observed phenotypic change in acceptor cells upon knockdown of single genes in donor cells can well be explained by altered expression profile of the donor cells, and subsequently EV loading (as EV reflect donor cells) and a completely different EV-induced phenotype in the recipient cell. However, based on these hallmark papers (which we think are still very important and insightful because they demonstrate the extraordinary role of EVs), the dogma in the field is that EV-induced effects are mediated by single (or a few) molecules loaded in EVs. With our manuscript, we try to change this dogma, and let the field realize that EVs contain many different biomolecules that can affect recipient cells.

Additional major issues:

1) The biggest problem of the manuscript is that there is no attempt to prevent EV transfer, or transfer of a specific network of molecules between the cell lines. One would need to show that interfering with EV mediated transfer alters the phenotype of migration to make any conclusion on the importance of the transfer of molecules by EVs. Such experiment are of course very difficult to perform. In addition, the in vitro phenotype of transfer of Cre is rather subtle, so the phenotypic read-out is also difficult to work with. Additionally, although the authors show transfer of EV

associated membrane labeling dyes, there is no evidence that transfer of the proteins and RNA that they identify as part of EV actually occurs, not even in vitro.

Response:

We apologize for not describing in the introduction the current status of the literature sufficiently enough, since the functional transfer of EV-RNAs is extensively studied and proven by other groups. In an early landmark study by the Lotvall lab, it was shown that functional mRNA loaded in EVs released from mouse cells can be taken up by human mast cells and importantly that the uptake of this mRNA leads to translation into functional proteins (Valadi et al. 2007, Nature Cell Biology). Furthermore, multiple studies have shown that EV-loaded miRNAs can suppress their target genes in recipient cells (for example Hergenreider, 2012, Nature cell biology, Fong 2015, Nature Cell biology, Zhang et al. 2015, Nature). Additionally, another reporter system that relies on RNA transfer by EVs shows functional transfer of EV associated GlucB mRNA (Lai, et al., 2015, Nature Communications). We showed in human xenografts (Zomer et al., Cell 2015; Zomer et al., Nat Prot, 2016) and now in this manuscript in a mouse allograft system that Cre activity (mostly due to mRNA, see Zomer et al, Cell 2015) is functionally transferred. In the revised manuscript, we have adapted the introduction and better described these important hallmark findings to illustrate all the evidence in the literature that shows the functionality of EV-associated (m)RNA upon EV transfer. At page 3 and 4 of the introduction, we have added: *“EV-associated biomolecules such as EV-RNA are stable in EVs and functional upon delivery into recipient cells. For example upon EV uptake, vesicular mRNA is translated into functional proteins (Valadi, Ekstrom et al., 2007), and vesicular miRNAs suppresses target genes in recipient cells (Fong, Zhou et al., 2015, Hergenreider, Heydt et al., 2012, Zhang, Zhang et al., 2015). Moreover, EV RNA-based reporter systems have confirmed the transport of functional mRNA for Cre (Ridder, Sevko et al., 2015, Zomer, Maynard et al., 2015, Zomer, Steenbeek et al., 2016) or GlucB (Lai, Kim et al., 2015) into recipient cells.”*

We also agree with the other point of this reviewer that it would be informative to interfere with the EV mediated transfer, since rendering the donor cells vesiculation-deficient enables to study how recipient cells behave once they no longer take up EVs. As we explained above, we take an alternative approach which provides exactly the same information: studying recipient cells that do not take up EVs (which would be the same as recipient cells that do not take up EVs upon interference with the release of EVs of donor cells). In our experiments, we ‘mark’ this population of cells with DsRed, serving as an internal control that is requested by this reviewer. Importantly, our approach has the advantage that it does not necessarily require manipulation of the donor cells to distinguish between recipient cells that have, or have not, taken up EVs, and therefore we can exclude potential artifacts induced by donor cell manipulation. Since we compare the behavior between cells that do and do not take up EVs, our data provides exactly the information that this reviewer suggests. We thank this reviewer for pointing out that we did not well describe the advantage of our technology and in the revised manuscript we have better explained that the Cre-Lox system provides similar information to interfering with EV mediated transfer. On page 7 of the revised manuscript we have added: *“Moreover, the DsRed⁺ cells serve as an internal control for cells that have not taken up EVs, providing a similar control to inferring with EV mediated transfer by e.g. using vesiculation-deficient EV donor cells.”* Additionally, on page 15 we have added: *“we considered to study whether inhibition of the release of EVs by B16F10 cells would affect the migratory behavior of B16F1 and B16F10 recipient cells. Unfortunately, good tools to only inhibit EV release without affecting the donor cells do currently not exist. However, as mentioned above, the Cre-LoxP system allows to address exactly this question using an alternative approach: the DsRed⁺ cells did not take up EVs and will behave similarly to cells that did not take up EVs upon inhibition of EV release in the donor cells, and can therefore act as a control for cells that do take up EVs (i.e. GFP expressing cells)”*

2) The manuscript contains no convincing evidence that EVs transfer proteins or RNA between cancer cell populations in vivo. The Cre-lox based tracking system is clever, but since there is no data documenting that Cre protein or mRNA is packed into EVs - and only can be transferred between cells through EVs - the method is not conclusive proof that transfer of molecules between cell lines occur through EVs. It is concerning that when the Cre-donor cell line is cultured with the reporter line in vitro then there is almost no transfer of Cre - even though release and uptake of EVs is convincingly shown under these conditions. Also concerning is the more efficient transfer of Cre between the cell lines in vivo. That could be because EV transfer is more efficient in vivo, as suggested by the authors, but there is no data to support this and Cre might just as well be

transferred by other means than EVs. In addition, there are no controls to show that there is no spontaneous color switch in the reporter cells in culture or when these are injected into animals without the Cre⁺ cells. Figure 6 shows that transfer of Cre affects the migration of the B16F1 cells, but it doesn't show that Cre is transferred by EV, or that the altered migration has anything to do with transfer of a network of molecules. In addition, there are no controls with measurement of migration of DsRed⁺ B16F1 cells or B16F10 cells in "non-co-injected" tumors.

Response:

As explained at our answer to major issue 1, the functional transfer of EV cargo is extensively studied and proven by others and we have included a better overview in the Results and discussion section of the revised manuscript. Moreover, in two papers (the initial Zomer et al, Cell, 2015 and an extensive protocol with all details Zomer et al, Nat Prot 2016) we have published many lines of evidence and controls showing that the transfer of Cre activity is mediated by EVs, including "non-co-injected" recipient tumors. Below, we discuss a few of those control experiments performed in the aforementioned papers, and how we refer to these in our current paper:

First, we have shown that mRNA of Cre was present in EVs (Below, Zomer et al, 2015 Figure 2F). Interestingly, Cre protein was not detected in EVs by western blot (Below, Zomer et al. Figure 2E) showing that either Cre protein is not present in EVs, or present below the detection level of western blot.

Zomer et al., Cell, 2015. PMID: 26000481 Figure 2.

To test whether cells report the uptake of Cre mRNA-containing EVs, we intratumorally injected EVs produced by Cre cells into reporter⁺ tumors (below, Zomer et al. Figure 4B). Indeed, tumor reporter⁺ cells start to report the uptake of the Cre-EVs as seen by the appearance of GFP⁺ cells, while in the non-injected tumors, no GFP⁺ cells appeared.

To exclude that potential transfer of non-EV associated Cre, we injected recombinant non-EV Cre protein and lysate of Cre⁺ tumor cells, and this did not result in eGFP⁺ reporter⁺ cells, showing that Cre activity needs to be packaged into EVs for functional uptake by reporter⁺ cells (below, Zomer et al. Figure 4 C, D). Additionally, we did not observe background recombination of reporter⁺ cells in the Cell paper (below, Zomer et al. Figure B, C, D).

Zomer et al., Cell, 2015. PMID: 26000481 Figure 4B-D.

Next, we have addressed whether Cre activity can be transferred by other means, e.g. by gap junctions during cell-cell contact. First, we have shown that transfer of Cre⁺ activity can be transferred independent of cell-cell contact in a transwell assay (Below, Zomer et al. Figure 3D) and between physical separated tumors that exclusively consists of either Cre⁺ or reporter⁺ cells (Below, Zomer et al Figure S3).

Zomer et al., Cell, 2015. PMID: 26000481 Figure 3 D.

Zomer et al, Cell, 2015, PMID: 26000481 Figure S3.

Next, we excluded cell-cell fusion *in vitro* and *in vivo*. Cell fusion of CFP-Cre⁺ cells and reporter⁺ cells would lead to fused cells that expresses both GFP and CFP. This was not the case, which excludes that Cre transfer is mediated by cell fusion (Below, Zomer et al. Figure 3C and S2).

Zomer et al., Cell, 2015. PMID: 26000481 Figure 3 C.

Zomer et al., Cell, 2015. PMID: 26000481 Figure S2.

Together, these lines of evidence and controls demonstrate that the transfer of Cre activity is mediated by EVs. To make this clear to the readers of our manuscript we have added to the introduction at page 7: “We have previously shown that Cre is transferred between cells independent of cell-cell fusion and cell-cell contacts, and that Cre RNA is loaded into EVs (Zomer et al., 2015, Zomer et al., 2016). Moreover, reporter cells report the uptake of Cre (DsRed to eGFP switch) upon exposure to purified Cre⁺ EVs, but not upon exposure to Cre protein or lysed Cre⁺ cells (Zomer et al., 2015, Zomer et al., 2016)”.

3) *The analysis of the contents of the EVs comes across as forced, almost biased, towards showing a connection to migration, including using things like the "reciprocal ratio of abundance". It is not obvious (e.g., in Fig. 3) that regulators of migration are specifically enriched in the EVs. It just looks like there are general differences. What statistical evidence is there that EVs contain regulators of migration - as opposed to other pathways?*

Response:

This reviewer is correct that we have analyzed the connection of EV cargo to migration, and we would like to elaborate on our reasons for this. Using *in vivo* imaging we found that B16F1 and B16F10 have differential capacity to migrate (Figure 1A-D of our manuscript). This is the primary reason why we have analyzed whether the EVs that these cells release have the potential to transfer cargo that can influence migration of the recipient cells. Gene ontology analysis indeed showed that the cargo of EVs (Figure 4, Figure EV4 and Figure EV5) is statistically associated to numerous different biological processes, of which many are directly or indirectly linked to cell migration. In these figures, we highlighted the direct migration-linked biological processes in red (e.g. cell migration, wound healing, cell motility) and we highlighted the potential indirect migration-linked processes that can strengthen the phenotype in green (e.g. RNA processing, cell surface receptor signaling pathway, glycoprotein metabolic process). Unfortunately, due to limited space, and because we only study the phenotypic effect of cancer cell migration, we could not focus on all biological processes that we identified in our RNA sequencing and mass-spectrometry experiments. We completely agree with the reviewer that the other biological processes that we identified in our tumor-derived EVs are very interesting, but we consider this beyond the scope of our current manuscript, as we only study the effect of EV transfer on cancer cell migration. In the revised manuscript, we aim to stress to the reader that we perform the network-based analysis because the differential phenotype we have observed is cell migration, and not solely base this on the GO term analysis. To this end we added to the revised manuscript on page 11: “Since we observed differential capacity of migration of B16F1 and B16F10 (Figure 1A-D), we analyzed whether EV cargo has the potential to influence migratory processes.” And on page 13: “Because B16F1 and B16F10 cells have differential migration capacity, we analyzed whether cargo that is specifically loaded in EVs can affect migration.” Moreover, on page 14: “Since we found that B16F1 and B16F10 cancer cells have differential migratory capacity, we tested whether the EV cargo with the most theoretical weight can influence migration.”

Moreover, we now realize that it may help to get the general findings of our study better across to the reader by discussing this data more prominently in the manuscript. In the results section on page 11 of our manuscript, we have added a generalized description: “Gene ontology analysis on EV RNA and protein showed that the cargo of EVs is associated to many different biological processes including some that can be directly linked to cell migration such as cytoskeleton organization, mesenchymal cell development, mesoderm morphogenesis and response to axon injury (Figure EV4B). In addition to processes directly linked to migration, we found many processes that can more indirectly influence migratory capacity of a cell, such as regulation of microtubule polymerization, but also to RNA processing and epigenetic and posttranscriptional regulation of gene expression (Figure EV4B)”

In the results section on page 13, we have added a generalized description of our findings and added: “GO term analysis showed that the EV-enriched proteins and RNAs are involved in a variety of biological processes (Figure 4B). The top GO terms are related to processes known to be upregulated during metastases including cell migration, wound healing and morphogenesis (Figure 4B, highlighted in red). The highest functional enrichment is observed for cell surface receptor

signaling pathway in the 100K EVs (Figure 4B). Previously, other groups have shown that EV-associated RTKs can be functionally incorporated in the plasma membrane of recipient cells (Al-Nedawi, Meehan et al., 2008, Zhang, Deng et al., 2017). Moreover, >100 proteins that are linked to cell surface receptor signaling pathways were found to be enriched in EVs (Figure 4B, see also Appendix Table S6), implying that EVs could transfer receptor-mediated oncogenic signaling to other cells.”

In the concluding remarks section on page 17, we have added: “In addition to networks of migration-related RNA and protein, we also found that EVs carry biomolecules with diverse other functions. For instance, EVs of the B16F10 model are enriched for epigenetic and posttranscriptional regulation of gene expression, as well as (m)RNA processing, potentially bringing about diverse effects in cells that receive these EVs (Figure EV4B). Moreover, RNA molecules related to glycolysis (Figure 4 and Figure EV5; glycoprotein metabolic process) are enriched in EVs. Interestingly, glycolysis have been shown to be upregulated in migratory cancer cells (Han, Kang et al., 2013, Shiraishi, Verdone et al., 2015). Additionally, enrichment of receptor-mediated signaling in EVs (cell surface signaling pathway, Figure 4B, Figure EV5B and Appendix Table S6) may help to transfer oncogenic properties between cancer cells. Together, the transfer of molecules involved in the above-mentioned processes may further amplify the effectiveness of the transferred biomolecules involved in migration.”

4) Are there really 1000 different proteins and 12,000 RNA species packed into EVs? These numbers suggest that the selection of content of EVs is very unspecific. Is it even possible that one EV particle can contain that many different proteins and RNAs? Or do these numbers reflect that different EV particles contain only a subset of the proteins and RNAs. If so, how does that affect the idea of transfer of a network of regulators? With the proteomics data in hand, the authors appear to have an excellent data source to determine whether EVs mostly carry peptide degradation products or full length proteins. This is a major question in the field - and degradation products could have very interesting functions irrespective of the function of the native proteins.

Response:

Just based on size, individual EV can never carry all the different proteins and RNA molecules and we have added this to the concluding remarks on page 17 and 18: “As single EVs cannot contain all cell-expressed RNAs and proteins due to volume restrictions (Sverdlov, 2012...)”. Therefore, we profile the total EV content present in the tumor interstitial fluid, and study how the network of regulators can be transferred by the pool of EVs, and not single EVs.

In the literature, several mechanisms of selection of content of EVs have been described. For example, it has been postulated that the cargo of exosomes (EVs that arise from MVB fusion with the limiting plasma membrane) can be specially targeted. However, it has been shown that other EVs that bud directly from the plasma membrane contain a cargo that is less specific. Indeed, we observed more specific RNA and protein loading in the 100K EV fraction than the 16.5K EV fraction, which is in line with what has been suggested in the field (Figure 4B) (e.g. Haraszti et al., 2016). Therefore, as was previously suggested, we think that both active selective sorting and unspecific loading takes place.

The question is whether the fact that not every EV contains all proteins and RNA species will affect the idea of transfer of a network of regulators. Even if the recipient cells do not receive all biomolecules or only a small amount of these molecules, many different nodes within the pathways will be amplified, leading to an increased output of the signaling pathway. As shown by others (e.g. Condeelis et al, 2005; Wyckoff et al, 2007), amplifying multiple nodes in the same pathway will have two effects: first, the output of the pathway can be much stronger when two nodes are hit simultaneously, and second, not every recipient cell needs to receive the same cargo (which may vary between single vesicles) to have the same signaling network amplification. In the revised manuscript, we explain this at page 18: “In order to become motile in the primary tumor, cancer cells upregulate multiple parallel networks of signaling pathways involved in migration (Figure 3B, C) (Wang, Goswami et al., 2004, Wang, Goswami et al., 2005, Wyckoff, Wang et al., 2007). Importantly, analysis of migratory cells isolated from tumors with different genetic origins showed that upregulation of different genes in the same pathway can be altered in different cells to achieve the same migratory behavior (Condeelis, Singer et al., 2005, Wyckoff et al., 2007). Our data shows that the pool of EVs released by tumor cells transfer networks of interconnected RNAs and proteins.

These data illustrate that it may not be the upregulation of the level of one particular RNA or protein that mediates the phenotypic change of recipient cells, but it could be the upregulation of the activity of migratory pathways as a whole. The consequence of this finding is that the output in the recipient cells will be very robust; even if the recipient cells do not receive all biomolecules or only a small amount of these molecules as can be transported by EVs, the concerted action of the network will lead to the same migratory output of the recipient cell.“

We would like to thank this reviewer to suggest to study whether the EVs that we isolated mostly carry peptide degradation products or full length proteins. To do this, we looked into the individual gel band data to infer broad molecular weight information of observed versus expected/calculated molecular weight. Interestingly, only a small number of proteins (3.1% and 1.8% of respectively 16.5 and 100K E protein cargo) appeared to be comprised of truncated proteins not identified as degraded in the cell lysate. We have added this data to Appendix Table S1 and describe this data at page 9 of the revised manuscript: *“To test if the EVs contained truncated proteins, we analyzed the low molecular weight gel bands of the EV samples that contain small proteins and potentially truncated proteins (gel band 4 and 5 in Figure EV1 and Appendix Figure S1). This analysis showed that only 3.1% of the 16.5K EV protein cargo and 1.8% of the 100K EV protein cargo is comprised of truncated proteins not identified as degraded in the cell lysate (Appendix Table S1).”* In the materials and methods section on page 27, we have included the corresponding method for this analysis: *“Identification of degradation products in EVs. For each sample type (cell lysate, 16.5K EVs, 100K EVs) protein data was exported for each individual gel block: fraction 1-1.5, 1.5-2.5, 2.5-3.5, 3.5-4.5 and 4.5-5. For each gel block, the predicted molecular weight of all identified proteins was plotted. Next, to identify outliers in the lower molecular weight blocks, proteins were identified that have a predicted molecular weight ≥ 2 x the standard deviation of gel block 4 and 5 (fraction 3.5-5), which resulted in a cutoff of 71.35 kDa. Utilizing these criteria, outlier proteins were identified in cell lysate (22), 16.5K EVs (104) and 100K EVs (63), with a portion of the outlier proteins identified as outliers in EVs and not in the cell lysate (16.5K EVs: 98 (3.1% of 16.5K EV-identified proteins), 100K EVs: 59 (1.8% of 100K EV-identified proteins)).”*

5) The results of the analysis of the effects of co-injection of B16F1 and B16F10 cell lines in the one mouse that developed metastasis from the B16F1 cells are over-stated - when only one mouse out of 11 develops metastasis after co-injection, there is no statistical support for the idea that co-injection is important and the analysis of the resulting metastasis impossible to interpret. In addition, an effect of co-injection wouldn't have to be through transfer of EVs. Finally, no data is presented on metastasis in mice only injected with B16F1 cells.

Response:

We completely agree with the reviewer and have also discussed extensively within the lab whether to include this data. Although we also had our doubts, we initially preferred to present the reader all the experimental data, rather than cherry pick the significant ones. It has not been our intention to overstate and we had the impression that we had carefully made our statement by explaining that only one mouse out of 11 mice developed metastatic disease. Power analysis suggests that we need to hundreds of mice to find out whether this finding is a significant result. Since this is beyond the scope of our manuscript, we agree with the reviewer and editor to omit this experiment and conclusion from the revised manuscript. We deleted Fig S6 and the sentence *“and in this mouse the eGFP⁺ B16F1 cells had a 6-fold higher metastatic potential than their DsRed⁺ counterparts (Figure S6)”*.

6) It is great that the authors isolated EVs from tumors and not just the tumor cell lines grown in vitro - but because of that, they cannot conclude that the characterized EVs originate from the cancer cells. They might very well also come from stromal cells. This point isn't even discussed.

Response:

We agree with the reviewer that in our approach to isolate EVs from the tumor interstitial fluid, we cannot exclude the co-isolation of EVs originating from stromal cells. As requested, we have acknowledged this in the revised manuscript, and have explained that the major contribution of EVs in our EV-preparation originates from cancer cells, but it will also contain a stromal component. In the results and discussion section of the manuscript, on page 7 and 8 we added: *“Although B16 tumors consist of both non-cancer cells and cancer cells that produce EVs, it is expected that the vast majority of EVs in tumors are produced by cancer cells. First, from cancer cell line and patient*

data, it is known that cancer cells release more EVs than non-transformed cells (Cesi, Walbrecht et al., 2016, Logozzi, De Milito et al., 2009),. Acidic pH (Parolini, Federici et al., 2009) and hypoxia (King, Michael et al., 2012), both common characteristics of solid tumors, are thought to contribute to this increased EV release. Second, dysregulation of exocytic pathways and changes in plasma membrane organization have also been suggested to enhance EV release by cancer cells (Minciacchi, Freeman et al., 2015a). Third, cancer cells also release large EVs with a diameter of 1-10 µm, often referred to as oncosomes (Di Vizio, Morello et al., 2012, Minciacchi, You et al., 2015b). Lastly, cancer cells form the vast majority of cells in B16F1 and B16F10 tumors, and therefore the vast majority of EVs present in B16 tumors are derived from cancer cells.”

2nd Editorial Decision

21 March 2018

Thank you for submitting your revised manuscript for consideration by The EMBO Journal, and your patience with our response, which got delayed due to delayed referee input. Your revised study was sent back to all three referees for re-evaluation. Please find their comments enclosed below.

As you will see, both referees #1 and #3 remain overall more critical on the study, however we decided - in light of the strong support of the other referees - to give you the opportunity to revise your manuscript to address the referee's points.

In more detail, referee #2 finds that his/her concerns have been sufficiently addressed and is now broadly in favour of publication. However, while both referees #1 and #3 state that representation and interpretation of the proteomics/RNAseq data have been improved, referee #3 remains critical regarding essential controls for the new setting of sister B16 cell lines analysed in an immune competent context missing which in his/her view undermines the robustness of the phenotype transfer observed. In addition, referee #1 has persistent reservations that the major findings of your study are not well integrated and the functional relevance of the identified EV cargo remains under-explored.

While we usually only offer one single round of revision at The EMBO Journal, considering the positive comments of referee #2, we have decided to ask you to revise your manuscript regarding the points raised by referees #1 and #3. We agree with referee #3 that demonstration of robustness of the Cre/Lox transfer system in the current adjusted system will be critical.

Referee #1:

In this revised manuscript the authors have done a very good job clarifying the raised issues, which has resulted in a more careful interpretation of the data, and in a more balanced and better justified description of the claims. Yet one major issue remains in the dis-connect between the proteomics data and the function of EV-content on recipient cells: On the hand, the authors show that EVs induce cell migration in recipient cells, and on the other hand they show that EVs contain many hundreds of proteins and RNAs. The central point they now make is that EV-mediated trans-acting effect is mediated by multiple proteins ('nodes within networks') rather than single factors as shown for some proteins in earlier studies. Although very likely this is true, it remains highly unfulfilling that no functional role has been assigned to any of the RNAs/proteins contained within EVs. What is then the reason to generate these proteomic/transcriptome data in the first place? Sure - it has led to the insight that EV-proteomes represent multiple functionalities, but arguing which proteins are the real biological actors remains a theoretical exercise, as the authors acknowledge in their response and revised text. Without demonstrating which (or at least which group) of EV-proteins induce the downstream effects, for me this does not provide the mechanistic insight one would hope for. The statement in the response letter that the role of individual proteins 'can never be experimentally proven' is not the strongest defense: many approaches that were un-imaginable until recently are commonplace today. In the end, the manuscript would benefit from keeping the major findings (i.e. proteomics and biological effects of EVs) as separate entities, without artificially (or theoretically) connecting the two, simply because the data are lacking. This would avoid nebulous wording to

describe the conclusion, e.g. as in the abstract, stating that 'a diverse range of EV biomolecules ... concertedly amplify numerous nodes within signaling networks'. The 'diverse range of biomolecules' is too vague, and the 'amplification of nodes' just means that some proteins are present both in EVs and recipient cells. To me, this is all a very complicated way to say that EVs (which contain many proteins and RNAs) act on recipient cells by invoking a multi-faceted biological response (e.g. cell migration) - which is a fair conclusion doing justice to the data.

Referee #2:

This revised version of the article by Steenbeek et al answers my previous concerns to my satisfaction. I thank the authors for performing the requested experiment of transfer between B16F1 tumors, and am glad to see that the results confirm their hypothesis.

I am not entirely convinced, however, by some of the new sentences :

1) the new title is not very clear, the authors should have kept the original structure (Tumor cells that phenocopy migratory behavior *in vivo*) and just change the end to « exchange numerous nodes of signaling networks via EVs ». In the new title, it is not clear what « *in vivo* » relates to !

2) The sentence included p7-8 to answer the question on cellular origin of the isolated EVs should be shortened to highlight only the fact that tumor cells are the major component of the B16 tumors. The first argument on tumors secreting more EVs than other cells should be removed : despite claims in numerous reviews, I do not think the literature has convincingly shown that tumors secrete more EVs than other cells ! Some primary cells (fibroblasts, dendritic cells) release much more materials as EVs than most tumor cell lines : the amount of EV released is an intrinsic feature of cells, that is not correlated with tumorigenicity.

3) In the new sentence included p14 to explain better the reason for comparing abundance in EVs vs the cells : I understand and agree with the concept proposed, that a molecule should have an effect on a target cell only if its abundance in EV is much higher than in cells. I am not sure I agree, however, that the notion of « specific loading » or « specific sorting » of components into EVs is relevant to the point made here. A component can be very abundant in EVs just because it is very abundant in the cell cytosol, and passively transferred to EVs, thus not « specifically sorted ». However, it seems to me that a protein that is more strongly present in EVs than the secreting cells is necessarily specifically sorted to EVs. The sentence starting by « By contrast, ... not be specifically sorted to EVs... recipient cells » suggests the opposite (possible high enrichment in EVs vs cells, without specific sorting to EVs) : can the authors explain better their thinking, or amend this sentence ?

Referee #3:

The revisions to the text has greatly improved the manuscript, and the results are now much clearer presented and more carefully discussed. One major concern persist:

Major point

1) The manuscript still lacks the necessary controls required to state that the Cre/lox system can be used to faithfully track transfer of EVs between the cancer cell populations that were used IN THIS STUDY. The authors argue strongly that using new cell lines constitutes novelty over their previous work - but they must also be able and willing to present the controls that show that Cre/lox system works to track EVs in this new model system. The requested controls are neither experimentally nor conceptually difficult, but they are absolutely essential if the authors want to argue that color switch allows them to track which cells have taken up EVs and which have not. The controls presented to the reviewers are human breast cancer cell lines in immune compromised animals (and previously published data that the reviewer was aware of). It cannot be assumed that because Cre is packaged into EVs in the previously used system, it also will be packed into EVs of unrelated murine melanoma cell lines and mark transfer of EVs in immune competent mice - and be the only way of Cre transfer. THAT may well be, but controls are needed. If the requested controls cannot be provided, I urge that the data based on the tracking system are excluded from the paper - or at the

very, very least incredibly carefully discussed. If the system works in this cell line then the authors should be easily capable of showing at a minimum the presence of Cre mRNA in the EVs from tumor preparations as supportive evidence. That, combined with a better discussion of potential issues with the system would be acceptable (such as stating that other means of Cre transfer was not excluded in these cell lines but were not detected in the previous cell lines and under which conditions). In part, I insist on these controls because there is no explanation for the very low transfer of Cre in vitro, when release and uptake of EVs is convincingly shown under the same conditions using other means. Regardless of the difference between in vitro and in vivo, it suggests that Cre-induced color switch greatly underreports on which cells have taken up EVs - and therefore, argues that DsRed⁺ cells in the mixed tumors cannot be assumed to be non-EV uptaking cells. Similarly, controls showing the amount of color switching happening in the absence of a report cell line, in vivo, using the B16 cells used for this study and not T47D cells in a prior study should also be done for Fig. 1I. It is the scholarly thing to do if one wants to make arguments on transfer of EVs as mediators of increased cell migration.

Minor points:

- 1) The discussion of EVs isolated from tumors as being predominantly from cancer cells is not based on any data (e.g. FACS quantifying percentage of tumor cells versus all other cells), or even reference to estimates of the percentage of cancer cells in the tumors from the literature. The percentage of cancer cells is probably not larger than 70%, and the authors need to be more careful in their description and should provide at a minimum estimates from the literature.
- 2) The statement on p. 6, lines 1-2 is not clear: F1 and F10 cells were mixed and 6/9 mice had F10 cell metastasis while 1/11 mice had F1 cell metastases. Why are the mouse numbers not the same (9 vs. 11 mice), where these not coinjection, analyzed for both cell populations? Please describe the experiment and findings better.
- 3) p. 9, line 9: it is not clear to me why authors exclude truncated proteins identified as degraded in the cell lysate from the analysis of truncated proteins.

2nd Revision - authors' response

2 May 2018

Comments from the editor

As you will see, both referees #1 and #3 remain overall more critical on the study, however we decided - in light of the strong support of the other referees - to give you the opportunity to revise your manuscript to address the referee's points.

Response:

We thank the editor for the opportunity to revise our manuscript and are pleased to include additional control experiments to address the remaining referee's points that further strengthen our findings.

In more detail, referee #2 finds that his/her concerns have been sufficiently addressed and is now broadly in favour of publication. However, while both referees #1 and #3 state that representation and interpretation of the proteomics/RNAseq data have been improved, referee #3 remains critical regarding essential controls for the new setting of sister B16 cell lines analysed in an immune competent context missing which in his/her view undermines the robustness of the phenotype transfer observed.

Response:

We acknowledge the concerns of reviewer #3 and we have included the three requested control experiments in the revised manuscript.

First, we have added proof of presence of Cre mRNA in both 16.5K and 100K EVs isolated from tumor micro environmental EVs, demonstrating that cancer cells from this model also load the Cre enzyme into EVs. We have added this data to Figure 1F and at page 7 of the results section, we have added "In the EVs released by Cre-expressing B16 cells, the mRNA of Cre was present (Figure 1F)".

Second, we have excluded Cre transfer through cell-cell fusion by excluding that GFP⁺ reporter⁺ cells also express CFP, and in the revised manuscript have added this data to Figure EV1 and page 7 of the results section: “*Importantly, GFP⁺ cells did not express CFP, excluding that the reporter⁺ cells fused with Cre⁺ cells (Figure EV 1A-D)*”.

Third, we excluded spontaneous recombination and appearance of GFP cells by showing the lack of GFP⁺ cells in tumors consisting of only reporter⁺ cells. In the revised manuscript, we have added this data to Figure 1H, J and at page 7 of the results section we have added “*we observed reporter⁺ cells that report Cre-activity only in tumors that also contained Cre⁺ cells (Figure H, J)*”.

In addition, referee #1 has persistent reservations that the major findings of your study are not well integrated and the functional relevance of the identified EV cargo remains under-explored.

Response:

To avoid over-stating causalities between OMICs data on the EV cargo, EV transfer and cellular behavior in the absence of additional functional experiments, we have restructured the manuscript, and show in the first part (opposed to the last part in the initial manuscript) the consequence of EV uptake on migratory behavior. Therefore this manuscript is now divided into two parts that are connected but not necessarily causal (to avoid overstatements): 1) two sister clones phenocopying behavior in vivo, 2) the analysis of the cargo of the EVs that they release.

While we usually only offer one single round of revision at The EMBO Journal, considering the positive comments of referee #2, we have decided to ask you to revise your manuscript regarding the points raised by referees #1 and #3. We agree with referee #3 that demonstration of robustness of the Cre/Lox transfer system in the current adjusted system will be critical.

In the revised manuscript, we have included the requested control experiments for the Cre/Lox transfer system and are therefore confident on positive feedback.

Comments from the reviewers

Referee #1:

In this revised manuscript the authors have done a very good job clarifying the raised issues, which has resulted in a more careful interpretation of the data, and in a more balanced and better justified description of the claims.

Response:

We thank the referee for his/her time to review our revised manuscript. We are contented that we have been able to solve the reviewers concerns on the EV-mediated transfer of mRNA between cells and the character of the networks identified in the Mass-spectrometry and RNA sequencing data.

Yet one major issue remains in the dis-connect between the proteomics data and the function of EV-content on recipient cells: On the hand, the authors show that EVs induce cell migration in recipient cells, and on the other hand they show that EVs contain many hundreds of proteins and RNAs. The central point they now make is that EV-mediated trans-acting effect is mediated by multiple proteins ('nodes within networks') rather than single factors as shown for some proteins in earlier studies. Although very likely this is true, it remains highly unfulfilling that no functional role has been assigned to any of the RNAs/proteins contained within EVs. What is then the reason to generate these proteomic/transcriptome data in the first place? Sure - it has led to the insight that EV-proteomes represent multiple functionalities, but arguing which proteins are the real biological actors remains a theoretical exercise, as the authors acknowledge in their response and revised text. Without demonstrating which (or at least which group) of EV-proteins induce the downstream effects, for me this does not provide the mechanistic insight one would hope for. The statement in the response letter that the role of individual proteins 'can never be experimentally proven' is not the strongest defense: many approaches that were un-imaginable until recently are commonplace today. In the end, the manuscript would benefit from keeping the major findings (i.e. proteomics and biological effects of EVs) as separate entities, without artificially (or theoretically) connecting the two, simply because the data are lacking.

Response:

We appreciate that many approaches were un-imaginable until recently are commonplace today, but it is important to realize that the development of such an approach has been a major and unsuccessful effort in the field. Although we have no doubt that this may be solved at some point in time by the field, we strongly feel that this goes beyond the scope of our current manuscript. Instead, we followed the suggestion of this reviewer to keep the major findings (biological effects of EVs and proteomics) as two separate entities to avoid over-stating the causality of the identified EV cargo and the observed transfer of the migratory phenotype. In the first part of the revised manuscript, we now describe the biological effects. To do this, we have moved the last paragraph of the results (describing the intravital microscopy of the consequence of EV) to the front of the result section. From the first part of the manuscript, we conclude that sister clones can phenocopy migratory behavior. In the second part of the results, we describe all experiments in which we identify the cargo. With this order, we avoid any overstatements of the causality of the identified EV cargo and the observed transfer of the migratory phenotype.

This would avoid nebulous wording to describe the conclusion, e.g. as in the abstract, stating that 'a diverse range of EV biomolecules ... concertedly amplify numerous nodes within signaling networks'. The 'diverse range of biomolecules' is too vague, and the 'amplification of nodes' just means that some proteins are present both in EVs and recipient cells. To me, this is all a very complicated way to say that EVs (which contain many proteins and RNAs) act on recipient cells by invoking a multi-faceted biological response (e.g. cell migration) - which is a fair conclusion doing justice to the data.

Response:

Thanks for this suggestion how to correctly phrase the abstract. We have followed the suggestion of this reviewer and have restated the conclusion of the abstract to: "Thus EVs, which contain many proteins and RNAs, act on recipient cells by invoking a multi-faceted biological response including cell migration."

Referee #2:

This revised version of the article by Steenbeek et al answers my previous concerns to my satisfaction. I thank the authors for performing the requested experiment of transfer between B16F1 tumors, and am glad to see that the results confirm their hypothesis.

Response:

We thank the reviewer for his/her positive feedback and are pleased to hear that we have been able to solve the previous concerns.

I am not entirely convinced, however, by some of the new sentences :

1) the new title is not very clear, the authors should have kept the original structure (Tumor cells that phenocopy migratory behavior in vivo) and just change the end to « exchange numerous nodes of signaling networks via EVs ». In the new title, it is not clear what « in vivo » relates to !

Response:

We agree, and have changed the title in the revised manuscript as suggested.

2) The sentence included p7-8 to answer the question on cellular origin of the isolated EVs should be shortened to highlight only the fact that tumor cells are the major component of the B16 tumors. The first argument on tumors secreting more EVs than other cells should be removed : despite claims in numerous reviews, I do not think the literature has convincingly shown that tumors secrete more EVs than other cells ! Some primary cells (fibroblasts, dendritic cells) release much more materials as EVs than most tumor cell lines : the amount of EV released is an intrinsic feature of cells, that is not correlated with tumorigenicity.

Response:

Thanks for pointing this out. In the revised manuscript, we have removed these incorrect statements, and now only highlight that the cancer cells are the major component of B16 tumors.

3) *In the new sentence included p14 to explain better the reason for comparing abundance in EVs vs the cells : I understand and agree with the concept proposed, that a molecule should have an effect on a target cell only if its abundance in EV is much higher than in cells. I am not sure I agree, however, that the notion of « specific loading » or « specific sorting » of components into EV is relevant to the point made here. A component can be very abundant in EVs just because it is very abundant in the cell cytosol, and passively transferred to EVs, thus not « specifically sorted ». However, it seems to me that a protein that is more strongly present in EVs than the secreting cells is necessarily specifically sorted to EVs. The sentence starting by « By contrast, ... not be specifically sorted to EVs... recipient cells » suggests the opposite (possible high enrichment in EVs vs cells, without specific sorting to EVs) : can the authors explain better their thinking, or amend this sentence ?*

Response:

We apologize for unclear wording, and completely agree with this comment. In the revised manuscript (page 14), we have changed this sentence to: *As another example, biological molecules that are passively transferred to EVs can potentially have a large effect if these molecules are at low abundance in the recipient cell.*

Referee #3:

The revisions to the text has greatly improved the manuscript, and the results are now much clearer presented and more carefully discussed. One major concern persist:

Response:

We also thank this reviewer for his/her positive comments on our revised manuscript, and the overall support of publishing our work. Below we have addressed the last remaining points.

Major point

1) The manuscript still lacks the necessary controls required to state that the Cre/lox system can be used to faithfully track transfer of EVs between the cancer cell populations that were used IN THIS STUDY. The authors argue strongly that using new cell lines constitutes novelty over their previous work - but they must also be able and willing to present the controls that show that Cre/lox system works to track EVs in this new model system. The requested controls are neither experimentally nor conceptually difficult, but they are absolutely essential if the authors want to argue that color switch allows them to track which cells have taken up EVs and which have not. The controls presented to the reviewers are human breast cancer cell lines in immune compromised animals (and previously published data that the reviewer was aware of). It cannot be assumed that because Cre is packaged into EVs in the previously used system, it also will be packed into EVs of unrelated murine melanoma cell lines and mark transfer of EVs in immune competent mice - and be the only way of Cre transfer. That may well be, but controls are needed. If the requested controls cannot be provided, I urge that the data based on the tracking system are excluded from the paper - or at the very, very least incredibly carefully discussed.

Response:

We completely agree with this reviewer, and as explained below, in the revised manuscript, we have added the three requested control experiments to support that Cre transfer is mediated through EVs in this model (see below).

If the system works in this cell line then the authors should be easily capable of showing at a minimum the presence of Cre mRNA in the EVs from tumor preparations as supportive evidence.

Response:

In the revised manuscript in Figure 1F we now illustrate the presence of Cre mRNA in EVs from B16 tumor preparations, and describe this data at page 7 of the results section "*In the EVs released by Cre-expressing B16 cells, the mRNA of Cre was present (Figure 1F)*". This data illustrates that Cre mRNA is present in the *in vivo* released EVs by murine melanoma cancer cells.

That, combined with a better discussion of potential issues with the system would be acceptable (such as stating that other means of Cre transfer was not excluded in these cell lines but were not

detected in the previous cell lines and under which conditions). In part, I insist on these controls because there is no explanation for the very low transfer of Cre in vitro, when release and uptake of EVs is convincingly shown under the same conditions using other means.

Response:

We agree, and in the revised manuscript (page 7) we included the following statement: *“Nevertheless, formally we cannot exclude a small fraction of the Cre exchange may have occurred via EV-independent mechanisms, such as Cre transfer through cell-cell contacts. However, the latter one we excluded previously in other tumor models in which cells exchanged Cre activity when located in physically separated tumors (Zomer et al., 2015).*

Moreover, in the revised manuscript we have added two additional controls (Extended view Figure 1A-D and Figure 1H, J).

First, we excluded Cre transfer upon cell fusion by showing that GFP⁺ cells are negative for CFP. We have added this data in Extended View Figure 1. At page 7 of the results section, we have added *“Importantly, GFP⁺ cells did not express CFP excluding that the reporter⁺ cells fused with Cre⁺ cells (Figure EV 1A-D)”*.

Second, we exclude that the color switch happens in the absence of Cre⁺ cells (see below).

Regardless of the difference between in vitro and in vivo, it suggests that Cre-induced color switch greatly underreports on which cells have taken up EVs - and therefore, argues that DsRed⁺ cells in the mixed tumors cannot be assumed to be non-EV uptaking cells.

Response:

We fully agree with the reviewer that the uptake of EVs (i.e. uptake of labeled EVs in Figure 1E) is not the same as the functional uptake of luminal EV cargo as reported by the Cre-LoxP system. In fact, the uptake of labeled EVs is most likely just reflecting basic endocytosis where cells are not exposed to the luminal EV cargo. The strength of the Cre-LoxP system is that it only reports the functional release of the luminal EV cargo into the recipient cells, which is strikingly different in the *in vitro* and *in vivo* setting.

To clarify this difference in methodology and readout better to the reader, we have rephrased some of the methodology in the revised manuscript. On page 7 of the revised manuscript, we have added: *“Reporter cells that take up these EVs, and get exposed to the cargo, switch expression of DsRed to eGFP”* and *“This data suggest that the Cre-Lox system reports the release of cargo into the cytoplasm rather than only the uptake of EVs, and that the in vitro EV uptake (i.e. uptake of labeled EVs in Figure 1E) did not coincide with substantial functional release of the content (i.e. lack of Cre-mediated color switch in Figure 1 I, J).”*

Similarly, controls showing the amount of color switching happening in the absence of a report cell line, in vivo, using the B16 cells used for this study and not T47D cells in a prior study should also be done for Fig. 1I. It is the scholarly thing to do if one wants to make arguments on transfer of EVs as mediators of increased cell migration.

Response:

We agree and have included this analysis to the revised manuscript. In Figure 1 H and J we now show that B16F1 and B16F10 reporter⁺ cells do not display any GFP expression in the absence of Cre⁺ cells. We describe this data at page 7: *“Indeed, in tumors we observed reporter⁺ cells that report Cre-activity only in tumors that also contained Cre⁺ cells (Figure 1H, I)”*.

Minor points:

1) *The discussion of EVs isolated from tumors as being predominantly from cancer cells is not based on any data (e.g. FACS quantifying percentage of tumor cells versus all other cells), or even reference to estimates of the percentage of cancer cells in the tumors from the literature. The percentage of cancer cells is probably not larger than 70%, and the authors need to be more careful in their description and should provide at a minimum estimates from the literature.*

Response:

To quantify the percentage of cancer cells versus all other cells in the tumor, we performed immunohistochemical analysis that we included in Appendix Figure S1 of the revised manuscript. The estimation of this reviewer was correct; this analysis showed that 70% of the cells of the tumor were cancer cells. We describe this data on page 8 of the manuscript: “*Since B16 tumors consist predominately of cancer cells (on average >70%, Appendix Figure S1), it is expected that the vast majority of EVs in tumors are produced by cancer cells, although we cannot exclude the co-isolation of some EVs derived from non-cancer cells*”.

2) *The statement on p. 6, lines 1-2 is not clear: F1 and F10 cells were mixed and 6/9 mice had F10 cell metastasis while 1/11 mice had F1 cell metastases. Why are the mouse numbers not the same (9 vs. 11 mice), where these not coinjection, analyzed for both cell populations? Please describe the experiment and findings better.*

Response:

We apologize for being insufficiently clear about the number of mice analyzed for these experiments. We injected 20 mice with a mixture of B16F1 and B16F10 cells and the resulting tumors have different ratios of B16F1 and B16F10 cells. Therefore, we only assessed the metastatic potential of B16F1 or B16F10 cells if they form the majority of cells in the primary tumor. In other words, if a tumor consisted of 90% of B16F10 cells and 10% of B16F1 cells, we analyzed if the B16F10 cells metastasized, and vice versa. Accordingly, we analyzed 11 mice in which B16F1 cells compose the bulk of the tumor, and we analyzed 9 mice in which the B16F10 cells compose the bulk of the tumor. In the text, we have clarified this by adding to page 5 and 6: “*Subcutaneous injection of fluorescently labelled B16F1 and B16F10 cells in 20 C57BL/6 mice led to tumors within 3 to 4 weeks, and co-injection of both lines led to tumors that contain both fluorescent B16 sub clones. In 9 mice, B16F10 cells formed the majority (>50%) of the cancer cells while in 11 mice the majority was formed by B16F1 cancer cells. Examination of lungs, lymph nodes and livers showed the presence of micro-metastases derived from B16F10 cells (6 out of 9 mice in which B16F10 cells are the major cancer type) but only occasionally from B16F1 cells (1 out of 11 mice in which the B16F1 cells are the major cancer type) confirming their differential metastatic potential*”.

3) *p. 9, line 9: it is not clear to me why authors exclude truncated proteins identified as degraded in the cell lysate from the analysis of truncated proteins.*

Response:

We thanks this reviewer for pointing out that this is not clear. In the revised manuscript, we included all truncated proteins, which led to almost the same number (3.1% of the 16.5K EV protein cargo and 1.9% of the 100K EV protein cargo). We corrected this at page 9 of the revised manuscript: “*This analysis showed that only 3.2% of the 16.5K EV protein cargo and 1.9% of the 100K EV protein cargo is comprised of truncated proteins (Appendix Table S1)*” and in the materials and methods on page 28: “*Utilizing these criteria, outlier proteins were identified in cell lysate (22), 16.5K EVs (104:3.2% of 16.5K EV-identified proteins) and 100K EVs (63:1.9% of 100K EV-identified proteins, with a portion of the outlier proteins identified as outliers in EVs and not in the cell lysate (16.5K EVs: 98 (3.1% of 16.5K EV-identified proteins), 100K EVs: 59 (1.8% of 100K EV-identified proteins))*”.

Corresponding Author Names: Jacco van Rheenen and Connie R. Jimenez

Journal Submitted to: The EMBO Journal

Manuscript Number: EMBOJ-2017-98357R